# Cerebellar learning using perturbations

**Guy Bouvier[1†‡], Johnatan Aljadeff[2§], Claudia Clopath[3], Célian Bimbard[1#],
Jonas Ranft[1], Antonin Blot[1¶], Jean-Pierre Nadal[4,5], Nicolas Brunel[2**],
Vincent Hakim[4], Boris Barbour[1*]**

[1]Institut de biologie de l'École normale supérieure (IBENS), École normale
supérieure, CNRS, INSERM, PSL University, Paris, France; [2]Departments of Statistics
and Neurobiology, University of Chicago, Chicago, United States; [3]Department of
Bioengineering, Imperial College London, London, United Kingdom; [4]Laboratoire de
Physique Statistique, École normale supérieure, CNRS, PSL University, Sorbonne
Université, Paris, France; [5]Centre d'Analyse et de Mathématique Sociales, EHESS,
CNRS, PSL University, Paris, France

**\*For correspondence:**
boris.barbour@ens.fr

**Present address:** [†]Department
of Physiology, University of
California, San Francisco, San
Francisco, United States;
[‡]Sandler Neuroscience,
University of California, San
Francisco, San Francisco, United
States; [§]Department of
Bioengineering, Imperial College
London, London, United
Kingdom; [#]Laboratoire des
systèmes perceptifs,
Département d'études
cognitives, École normale
supérieure, PSL University, Paris,
France; [¶]Sainsbury-Wellcome
Centre for Neural Circuits and
Behaviour, University College
London, London, United
Kingdom; [**]Departments of
Neurobiology and Physics, Duke
University, Durham, United
States

**Competing interests:** The
authors declare that no
competing interests exist.

**Reviewing editor:** Jennifer L
Raymond, Stanford School of
Medicine, United States

**Abstract** The cerebellum aids the learning of fast, coordinated movements. According to
current consensus, erroneously active parallel fibre synapses are depressed by complex spikes
signalling movement errors. However, this theory cannot solve the *credit assignment problem* of
processing a global movement evaluation into multiple cell-specific error signals. We identify a
possible implementation of an algorithm solving this problem, whereby spontaneous complex
spikes perturb ongoing movements, create eligibility traces and signal error changes guiding
plasticity. Error changes are extracted by adaptively cancelling the average error. This framework,
*stochastic gradient descent with estimated global errors* (SGDEGE), predicts synaptic plasticity
rules that apparently contradict the current consensus but were supported by plasticity
experiments in slices from mice under conditions designed to be physiological, highlighting the
sensitivity of plasticity studies to experimental conditions. We analyse the algorithm's convergence
and capacity. Finally, we suggest SGDEGE may also operate in the basal ganglia.
DOI: https://doi.org/10.7554/eLife.31599.001

## Introduction

A central contribution of the cerebellum to motor control is thought to be the learning and automatic execution of fast, coordinated movements. Anatomically, the cerebellum consists of a convoluted, lobular cortex surrounding the cerebellar nuclei (*Figure 1A*, *Eccles et al., 1967*; *Ito, 1984*). The main input to the cerebellum is the heterogeneous mossy fibres, which convey multiple modalities of sensory, contextual and motor information. They excite both the cerebellar nuclei and the cerebellar cortex; in the cortex they synapse with the very abundant granule cells, whose axons, the parallel fibres, excite Purkinje cells. Purkinje cells constitute the sole output of the cerebellar cortex and project an inhibitory connection to the nuclei, which therefore combine a direct and a transformed mossy fibre input with opposite signs. The largest cell type in the nuclei, the projection neurone, sends excitatory axons to several motor effector systems, notably the motor cortex via the thalamus. Another nuclear cell type, the nucleo-olivary neurone, inhibits the inferior olive. The cerebellum receives a second external input: climbing fibres from the inferior olive, which form an extensive, ramified connection with the proximal dendrites of the Purkinje cell. Each Purkinje cell receives a single climbing fibre. A more modular diagram of the olivo-cerebellar connectivity relevant to this paper is shown in *Figure 1B*; numerous cell types and connections have been omitted for simplicity.

Purkinje cells discharge two distinct types of action potential (*Figure 1C*). They nearly continuously emit *simple spikes*—standard, if brief, action potentials—at frequencies that average 50 Hz. This frequency is modulated both positively and negatively by the intensity of inputs from the mossy

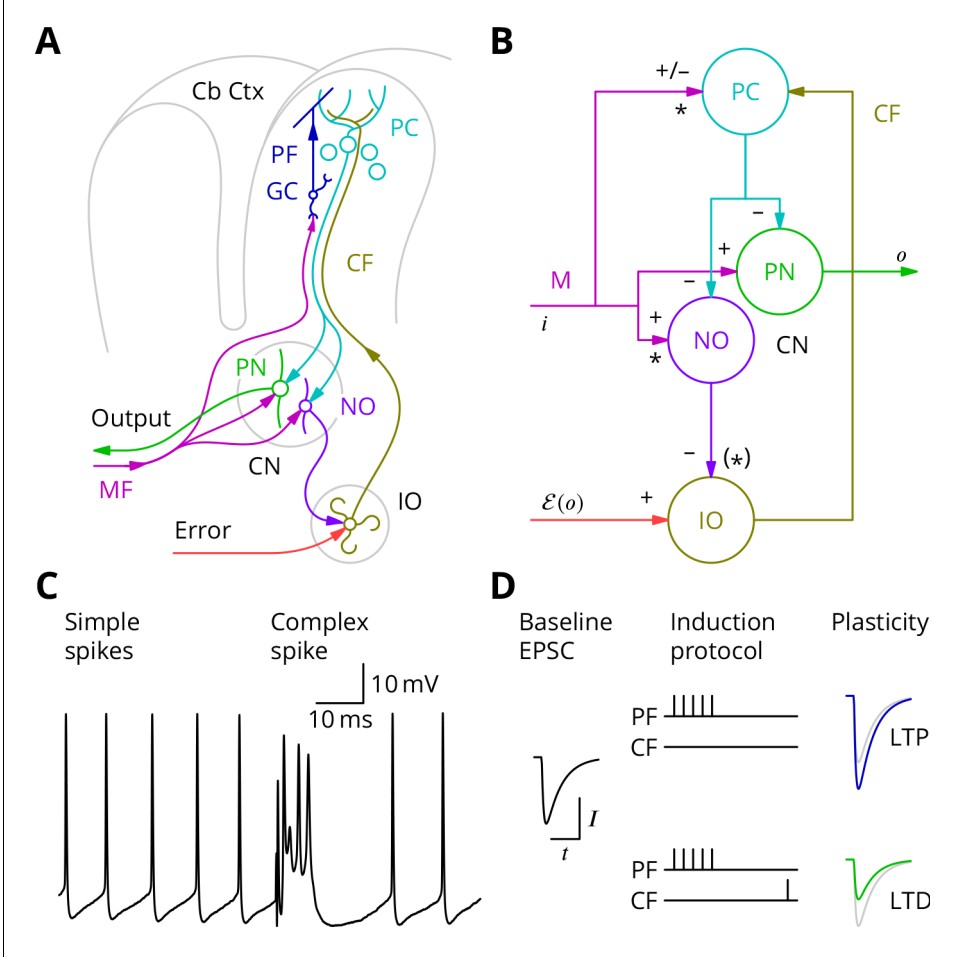

**Figure 1.** The cerebellar circuitry and properties of Purkinje cells. (A) Simplified circuit diagram. MF, mossy fibres; CN, (deep) cerebellar nuclei; GC, granule cells; Cb Ctx, cerebellar cortex; PF, parallel fibres; PC, Purkinje cells; PN, projection neurones; NO, nucleo-olivary neurones; IO, inferior olive; CF, Climbing fibres. (B) Modular diagram. The ± signs next to the synapses indicate whether they are excitatory or inhibitory. The granule cell and indirect inhibitory inputs they recruit have been subsumed into a bidirectional mossy fibre–Purkinje cell input, M. Potentially plastic inputs of interest here are denoted with an asterisk. $i$, input; $o$, output; $\mathcal{E}(o)$, error (which is a function of the output). (C) Typical Purkinje cell electrical activity from an intracellular patch-clamp recording. Purkinje cells fire two types of action potential: simple spikes and, in response to climbing fibre input, complex spikes. (D) According to the consensus plasticity rule, a complex spike will depress parallel fibre synapses active about 100 ms earlier. The diagram depicts idealised excitatory postsynaptic currents (EPSCs) before and after typical induction protocols inducing long-term potentiation (LTP) or depression (LTD). Grey, control EPSC; blue, green, post-induction EPSCs.

DOI: https://doi.org/10.7554/eLife.31599.002

fibre–granule cell pathway (which can also recruit interneurons that inhibit Purkinje cells; *Eccles et al., 1967*). Such modulations of Purkinje cell firing are thought to underlie their contributions to motor control. In addition, when the climbing fibre is active, an event that occurs continuously but in a somewhat irregular pattern with a mean frequency of around 1 Hz, the Purkinje cell emits a completely characteristic *complex spike* under the influence of the intense excitation from the climbing fibre (*Figure 1C*).

The history of research into cerebellar learning is dominated by the theory due to *Marr (1969)* and *Albus (1971)*. They suggested that the climbing fibre acts as a 'teacher' to guide plasticity of parallel fibre–Purkinje cell synapses. It was several years, however, before experimental support for this hypothesis was obtained (*Ito et al., 1982*; *Ito and Kano, 1982*), by which time the notion that

the climbing fibre signalled errors had emerged (*Ito, 1972*; *Ito, 1984*). Error modalities thought to be represented by climbing fibres include: pain, unexpected touch, imbalance, and retinal slip. According to the modern understanding of this theory, by signalling such movement errors, climbing fibres induce long-term depression (LTD) of parallel fibre synapses that were active at the same time (*Ito et al., 1982*; *Ito and Kano, 1982*; *Sakurai, 1987*; *Crepel and Jaillard, 1991*) or, more precisely, shortly before (*Wang et al., 2000*; *Sarkisov and Wang, 2008*; *Safo and Regehr, 2008*). A compensating long-term potentiation (LTP) is necessary to prevent synaptic saturation (*Lev-Ram et al., 2002*; *Lev-Ram et al., 2003*; *Coesmans et al., 2004*) and its induction is reported to follow high-frequency parallel fibre activity in the absence of complex spikes (*Jörntell and Ekerot, 2002*; *Bouvier et al., 2016*). Plasticity of parallel fibre synaptic currents according to these plasticity rules is diagrammed in *Figure 1D*.

Cerebellar learning with the Marr-Albus-Ito theory has mostly been considered, both experimentally and theoretically, at the level of single cells or of uniformly responding groups of cells learning a single stereotyped adjustment. Such predictable and constrained movements, exemplified by eye movements and simple reflexes, provide some of the best studied models of cerebellar learning: the vestibulo-ocular reflex (*Robinson, 1976*; *Ito et al., 1974*; *Blazquez et al., 2004*), nictitating membrane response/eye blink conditioning (*McCormick et al., 1982*; *Yeo et al., 1984*; *Yeo and Hesslow, 1998*), saccade adaptation (*Optican and Robinson, 1980*; *Dash and Thier, 2014*; *Soetedjo et al., 2008*) and regulation of limb movements by withdrawal reflexes (*Ekerot et al., 1995*; *Garwicz et al., 2002*). All of these motor behaviours have in common that there could conceivably be a fixed mapping between an error and a suitable corrective action. Thus, adaptations necessary to ensure gaze fixation are exactly determined by the retinal slip. Such fixed error-correction relations may have been exploited during evolution to create optimised correction circuitry.

The problems arise with the Marr-Albus-Ito theory if one tries to extend it to more complex situations, where neurones must respond heterogeneously (not uniformly) and/or where the flexibility to learn arbitrary responses is required. Many motor control tasks, for instance coordinated movements involving the hands, can be expected to fall into this class. To learn such complex/arbitrary movements with the Marr-Albus-Ito theory requires error signals that are specific for each cell, each movement and each time within each movement. The theory is thus incomplete, because it does not describe how a global evaluation of movement error can be processed to provide such detailed instructions for plasticity to large numbers of cells, a general difficulty which the brain has to face that was termed the *credit assignment problem* by *Minsky (1961)*.

A suitable algorithm for solving the general cerebellar learning problem would be *stochastic gradient descent*, a classical optimisation method that *Minsky (1961)* suggested might operate in the brain. We shall speak of 'descent' when minimising an error and 'ascent' when maximising a reward, but the processes are mathematically equivalent. In stochastic gradient descent, the objective function is explored by random variations in the network that alter behaviour, with plasticity then retaining those variations that improve the behaviour, as signalled by a decreased error or increased reward. Several possible mechanisms of varying biological plausibility have been proposed. In particular, perturbations caused by synaptic release (*Minsky, 1954*; *Seung, 2003*) or external inputs (*Doya and Sejnowski, 1988*) have been suggested, while various (abstract) mechanisms have been proposed for extraction of changes in the objective function (*Williams, 1992*). To avoid confusion, note that these forms of stochastic gradient descent differ from those conforming to the more restrictive definition used in the machine learning community, in which the stochastic element is the random sampling of examples from the training set (*Robbins and Monro, 1951*; *Shalev-Shwartz and Ben-David, 2014*). This latter form has achieved broad popularity in online learning from large data sets and in deep learning. Although the theoretical framework for (perturbative) stochastic gradient descent is well established, the goal of identifying in the brain a network and cellular implementation of such an algorithm has proved elusive.

The learning behaviour with the best established resemblance to stochastic gradient ascent is the acquisition of song in male songbirds. The juvenile song is refined by a trial and error process to approach a template memorised from a tutor during a critical period (*Konishi, 1965*; *Mooney, 2009*). The analogy with stochastic gradient ascent was made by *Doya and Sejnowski (1988)* and was then further developed experimentally (*Olveczky et al., 2005*) and theoretically (*Fiete et al., 2007*). However, despite these very suggestive behavioural correlates, relatively little

progress has been made in verifying model predictions for plasticity or identifying the structures responsible for storing the template and evaluating its match with the song.

A gradient descent mechanism for the cerebellum has been proposed by the group of *Dean et al. (2002)*, who term their algorithm *decorrelation*. Simply stated (*Dean and Porrill, 2014*), if both parallel fibre and climbing fibre inputs to a Purkinje cell are assumed to vary about their respective mean values, their correlation or anticorrelation indicates the local gradient of the error function and thus the sign of the required plasticity. At the optimum, which is a minimum of the error function, there should be no correlation between variations of the climbing fibre rate and those of the parallel fibre input. Hence the name: the algorithm aims to decorrelate parallel and climbing fibre variations. An appropriate plasticity rule for implementing decorrelation is a modified covariance rule (*Sejnowski, 1977*; who moreover suggested in abridged form a similar application to cerebellar learning). Although decorrelation provides a suitable framework, its proponents are still in the process of developing a cellular implementation (*Menzies et al., 2010*). Moreover, we believe that the detailed implementation suggested by *Menzies et al., 2010* is unable to solve a temporal credit assignment problem (that we identify below) arising in movements that can only be evaluated upon completion.

Below, we analyse in more detail how the Marr-Albus-Ito theory fails to solve the credit assignment problem and suggest how the cerebellum might implement stochastic gradient descent. We shall provide support for unexpected predictions the proposed implementation makes regarding cerebellar synaptic plasticity, which are different from those suggested for decorrelation. Finally, we shall perform a theoretical analysis demonstrating that learning with the algorithm converges and that it is able to attain maximal storage capacity.

## Results

### Requirements for cerebellar learning

We begin by examining the current consensus (Marr-Albus-Ito) theory of cerebellar learning and illustrating some of its limitations when extended to the optimisation of complex, arbitrary movements. The learning framework we consider is the following. The cerebellar circuitry must produce at its output trains of action potentials of varying frequencies at given times (*Thach, 1968*). We consider only firing rates $r(t)$, which are a function of time in the movement (*Figure 2A*). The cerebellar output influences the movement and the resulting movement error, which can be sensed and fed back to the Purkinje cells in the form of climbing fibre activity.

We constructed a simple network simulation to embody this framework (see Materials and methods). In it, mossy fibre drive to the cerebellum was considered to be movement- and time-dependent but not to vary during learning. Granule cells and molecular layer interneurons were collapsed into the mossy fibre inputs, which acted on Purkinje cells through unconstrained synapses (negative weights could arise through plasticity of interneuron synapses; *Jörntell and Ekerot, 2002*; *Jörntell and Ekerot, 2003*; *Mittmann and Häusser, 2007*) that were modified to optimise the firing profiles of the projection neurones of the deep cerebellar nuclei. We implemented the Marr-Albus-Ito algorithm via a plasticity rule in which synaptic weights were updated by the product of presynaptic activity and the presence of the error signal. Thus, active mossy fibre (pathway) inputs would undergo LTP (increasing Purkinje cell firing; *Lev-Ram et al., 2003*) after each movement unless the climbing fibre was active, in which case they would undergo LTD (decreasing activity). The model represented a microzone whose climbing fibres were activated uniformly by a global movement error.

The behaviour of the Marr-Albus-Ito algorithm in this network depends critically on how the definition of the error is extended from the error for an individual cell. The most natural definition for minimising all errors would be to sum the absolute differences between the actual firing profile of each cerebellar nuclear projection neurone and its target, see *Equation 7*. However, this error definition is incompatible with the Marr-Albus-Ito algorithm (*Unsigned* in *Figure 2B*). Examination in *Figure 2C* of the firing profile of one of the projection neurones shows that their rate simply saturates. The algorithm *is* able to minimise the average signed error defined in *Equation 14* (*Signed* in *Figure 2B*). However, inspection of a specimen final firing profile illustrates the obvious limitation

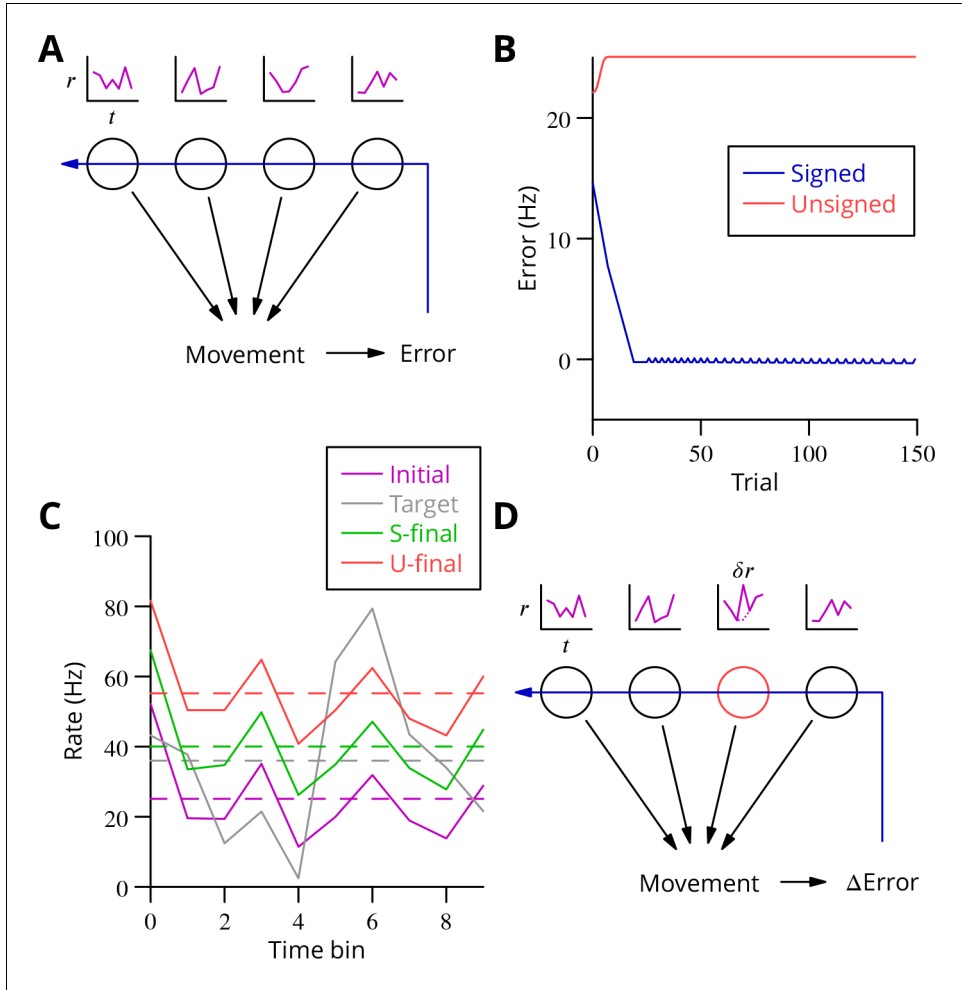

**Figure 2.** Analysis of cerebellar learning. (**A**) The model of cerebellar learning we address is to adjust the temporal firing profiles (miniature graphs of rate as a function of time, $r(t)$, *magenta*) of multiple cerebellar output neurones (nuclear projection neurones) (*black circles*) to optimise a movement, exploiting the evaluated movement error, which is fed back to the cerebellar circuit (Purkinje cells) (*blue arrow*). (**B**) The Marr-Albus-Ito theory was simulated in a network simulation embodying different error definitions (Results, Materials and methods). The Marr-Albus-Ito algorithm is able to minimise the average signed error (*blue*, *Signed*) of a group of projection neurones, but not the unsigned error (*red*, *Unsigned*). (**C**) However, as expected, optimising the signed error does not optimise individual cell firing profiles: comparison of the underlying firing profiles (initial, *magenta*; final, *green*) with their target (*grey*) for a specimen projection neurone illustrates that neither the temporal profile nor even the mean single-cell firing rate is optimised. If the unsigned error is used, there is no convergence and the firing rates simply saturate (*red*) and the error *increases*. (**D**) A different learning algorithm will be explored below: stochastic gradient descent, according to which temporally and spatially localised firing perturbations ($\delta r$, *red*, of a Purkinje cell) are consolidated if the movement improves; this requires extraction of the *change* of error ($\Delta$ Error).
DOI: https://doi.org/10.7554/eLife.31599.003

that neither the detail of the temporal profile nor even the mean firing rate for individual cells is optimised when using this error (see *Figure 2C*).

This limitation illustrates the credit assignment problem. How to work back from a single evaluation after the movement to optimise the firing rate of multiple cells at different time points, with all of those firing profiles also differing between multiple movements which must all be learnt in parallel? A model problem that the Marr-Albus-Ito algorithm cannot solve would be two neurones receiving the same inputs and error signal but needing to undergo opposite plasticity. In the general case of learning complex, arbitrary movements, this requires impractical foreknowledge to be embodied by the climbing fibre system, to know exactly which cell must be depressed to optimise each

movement, and this still would not solve the problem of inducing plasticity differentially at different time points during a firing profile, given that often only a single, post-movement evaluation will be available. An example would be a saccade, where the proximity to the target can only be assessed once the movement of the eye has ceased.

## The complex spike as trial and error

The apparent difficulty of solving the credit assignment problem within the Marr-Albus-Ito algorithm led us to consider whether the cerebellum might implement a form of stochastic gradient descent. By combining a known perturbation of one or a small group of neurones with information about the change of the global error (*Figure 2D*), it becomes possible to consolidate, specifically in perturbed neurones, those motor command modifications that reduce the error, thus leading to a progressive optimisation. We set out to identify a biologically plausible implementation in the cerebellum. In particular, that implementation should be able to optimise multiple movements in parallel, employing only feasible cellular computations—excitation, inhibition and thresholding.

A cerebellar implementation of stochastic gradient descent must include a source of perturbations of the Purkinje cell firing rate $\delta r$. The fact that Purkinje cells can contribute to different movements with arbitrary and unknown sequencing imposes an implementation constraint preventing simple-minded approaches like comparing movements performed twice in succession. We recall that we assume that no explicit information categorising or identifying movements is available to the Purkinje cell. It is therefore necessary that knowledge of both the presence and sign of $\delta r$ be available within the context of a single movement execution.

In practice, a number of different perturbation mechanisms can still satisfy these requirements. For instance, any binary signal would be suitable, since the sign of the perturbation with respect to its mean would be determined by the simple presence or absence of the signal. Several plausible mechanisms along these lines have been proposed, including external modulatory inputs (*Doya and Sejnowski, 1988*; *Fiete et al., 2007*), failures and successes of synaptic transmission (*Seung, 2003*) or the absence and presence of action potentials (*Xie and Seung, 2004*). However, none of these mechanisms has yet attracted experimental support at the cellular level.

In the cerebellar context, parallel fibre synaptic inputs are so numerous that the correlation between individual input variations and motor errors is likely to be extremely weak, whereas we seek a perturbation that is sufficiently salient to influence ongoing movement. Purkinje cell action potentials are also a poor candidate, because they are not back-propagated to parallel fibre synapses (*Stuart and Häusser, 1994*) and therefore probably cannot guide their plasticity, but the ability to establish a synaptic eligibility trace is required. Bistable firing behaviour of Purkinje cells (*Loewenstein et al., 2005*; *Yartsev et al., 2009*), with the down-state (or long pauses) representing a clear perturbation towards lower (zero) firing rates, is a perturbation candidate. However, exploratory plasticity experiments did not support this hypothesis and the existence of bistability in vivo is disputed (*Schonewille et al., 2006a*).

We thus propose, in accordance with a suggestion due to *Harris (1998)*, another possible perturbation of Purkinje cell firing: the complex spike triggered by the climbing fibre. We note that there are probably two types of inferior olivary activity. Olivary neurones mediate classical error signalling triggered by external synaptic input, but they also exhibit continuous and irregular spontaneous activity in the absence of overt errors. We suggest the spontaneous climbing fibre activations cause synchronised *perturbation complex spikes* (pCSs) in small groups of Purkinje cells via the $\sim$ 1:10 inferior olivary–Purkinje cell divergence (*Schild, 1970*; *Mlonyeni, 1973*; *Caddy and Biscoe, 1976*), dynamic synchronisation of olivary neurones through electrical coupling (*Llinás and Yarom, 1986*; *Bazzigaluppi et al., 2012a*) and common synaptic drive. The excitatory perturbation—a brief increase of firing rate (*Ito and Simpson, 1971*; *Campbell and Hesslow, 1986*; *Khaliq and Raman, 2005*; *Monsivais et al., 2005*)—feeds through the cerebellar nuclei (changing sign; *Bengtsson et al., 2011*) to the ongoing motor command and causes a perturbation of the movement, which in turn may modify the error of the movement.

The perturbations are proposed to guide learning in the following manner. If a perturbation complex spike results in an increase of the error, the raised activity of the perturbed Purkinje cells was a mistake and reduced activity would be preferable; parallel fibre synapses active at the time of the perturbing complex spikes should therefore be depressed. Conversely, if the perturbation leads to a

reduction of error (or does not increase it), the increased firing rate should be consolidated by potentiation of the simultaneously active parallel fibres.

How could an increase of the error following a perturbation be signalled to the Purkinje cell? We suggest that the climbing fibre also performs this function. Specifically, if the perturbation complex spike increases the movement error, a secondary *error complex spike* (eCS) is emitted shortly afterwards, on a time scale of the order of 100 ms (50–300 ms). This time scale is assumed because it corresponds to the classical error signalling function of the climbing fibre, because it allows sufficient time for feedback via the error modalities known to elicit complex spikes (touch, pain, balance, vision) and because such intervals are known to be effective in plasticity protocols (*Wang et al., 2000*; *Sarkisov and Wang, 2008*; *Safo and Regehr, 2008*). The interval could also be influenced by the oscillatory properties of olivary neurones (*Llinás and Yarom, 1986*; *Bazzigaluppi et al., 2012b*).

The predicted plasticity rule is therefore as diagrammed in *Figure 3*. Only granule cell synapses active simultaneously with the perturbation complex spike undergo plasticity, with the sign of the plasticity being determined by the presence or absence of a subsequent error complex spike. Granule cell synapses active in the absence of a synchronous perturbation complex spike should not undergo plasticity, even if succeeded by an error complex spike. We refer to these different protocols with the abbreviations (and give our predicted outcome in parenthesis): G_ _ (no change), GP_ (LTP), G_E (no change), GPE (LTD), where G indicates granule cell activity, P the presence of a perturbation complex spike and E the presence of an error complex spike. Note that both granule cells and climbing fibres are likely to be active in high-frequency bursts rather than the single activations idealised in *Figure 3*.

Several of the predictions of this rule appear to be incompatible with the current consensus. Thus, parallel fibre synapses whose activity is simultaneous with (GP_) or followed by a complex spike (G_E) have been reported to be depressed (*Sakurai, 1987*; *Crepel and Jaillard, 1991*; *Lev-Ram et al., 1995*; *Coesmans et al., 2004*; *Safo and Regehr, 2008*; *Gutierrez-Castellanos et al.,*

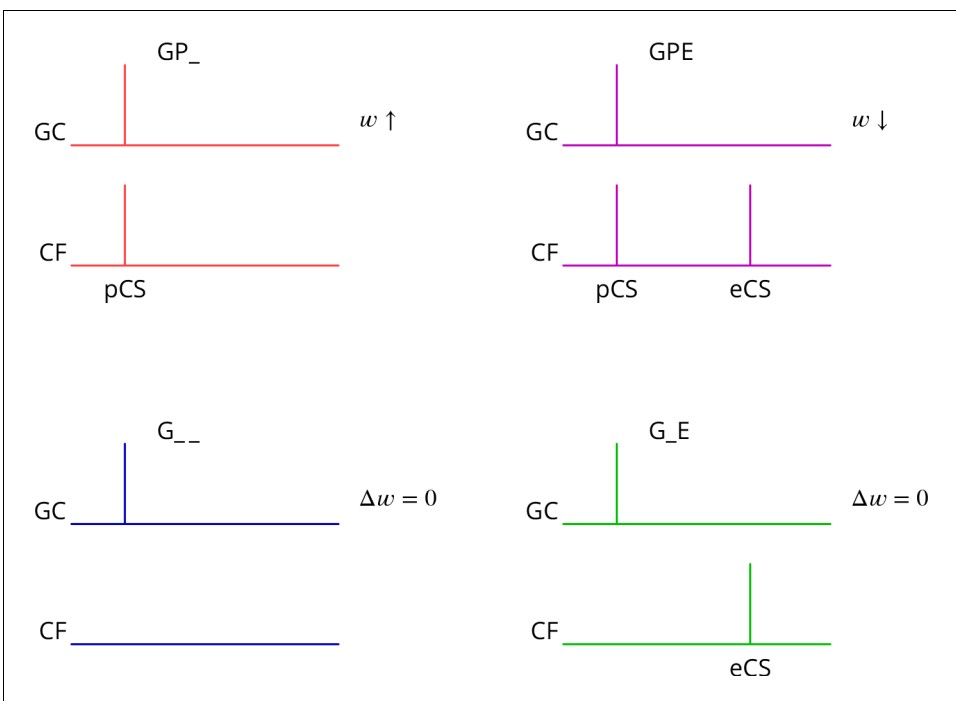

**Figure 3.** Predicted plasticity rules. Synchronous activation of granule cell synapses and a perturbation complex spike (pCS) leads to LTP (GP_, increased synaptic weight *w*; top left, *red*), while the addition of a succeeding error complex spike (eCS) leads to LTD (GPE, top right, *magenta*). The bottom row illustrates the corresponding 'control' cases from which the perturbation complex spike is absent; no plasticity should result (G_ _ *blue* and G_E *green*).
DOI: https://doi.org/10.7554/eLife.31599.004

2017), while we predict potentiation and no change, respectively. Furthermore, parallel fibre activity alone (G_ _) leads to potentiation (*Lev-Ram et al., 2002*; *Jörntell and Ekerot, 2002*; *Lev-Ram et al., 2003*; *Coesmans et al., 2004*; *Gutierrez-Castellanos et al., 2017*), while we predict no change.

## Synaptic plasticity under physiological conditions

As described above, the plasticity rules we predict for parallel fibre–Purkinje cell synapses are, superficially at least, close to the opposite of the consensus in the literature. Current understanding of the conditions for inducing plasticity gives a key role to the intracellular calcium concentration (combined with nitric oxide signalling; *Coesmans et al., 2004*; *Bouvier et al., 2016*), whereby high intracellular calcium concentrations are required for LTD and moderate concentrations lead to LTP. Standard experimental conditions for studying plasticity in vitro, notably the extracellular concentration of calcium, are likely to result in more elevated intracellular calcium concentrations during induction than pertain physiologically. Recognising that this could alter plasticity outcomes, we set out to test whether our predicted plasticity rules might be verified under more physiological conditions.

We made several changes to standard protocols (see Materials and methods); one was cerebellum-specific, but the others also apply to in vitro plasticity studies in other brain regions. We did not block GABAergic inhibition. We lowered the extracellular calcium concentration from the standard 2 mM (or higher) used in slice work to 1.5 mM (*Pugh and Raman, 2008*), which is near the maximum values measured in vivo in rodents (*Nicholson et al., 1978*; *Jones and Keep, 1988*; *Silver and Erecińska, 1990*). To avoid the compact bundles of active parallel fibres produced by the usual stimulation in the molecular layer, we instead used weak granule cell layer stimuli, which results in sparse and spatially dispersed parallel fibre activity. Interestingly, it has been reported that standard protocols using granule cell stimulation are unable to induce LTD (*Marcaggi and Attwell, 2007*). We used a pipette solution designed to prolong energy supply in extended cells like the Purkinje cell (see Materials and methods). Experiments were carried out in adult mouse sagittal cerebellar slices using otherwise standard patch-clamp techniques.

Pairs of granule cell test stimuli with an interval of 50 ms were applied at 0.1 Hz before and after induction; EPSCs were recorded in voltage clamp at −70 mV. Pairs of climbing fibre stimuli with a 2.5 ms interval were applied at 0.5 Hz throughout the test periods, mimicking tonic climbing fibre activity, albeit at a slightly lower rate. The interleaved test granule cell stimulations were sequenced 0.5 s before the climbing fibre stimulations. The analysis inclusion criteria and amplitude measurement for the EPSCs are detailed in the Materials and methods. The average amplitude of the granule cell EPSCs retained for analysis was −62 ± 46 pA (mean ± s.d., $n$ = 58). The rise and decay time constants (of the global averages) were 0.74 ± 0.36 ms and 7.2 ± 2.7 ms (mean ± sd), respectively.

During induction, performed in current clamp without any injected current, the granule cell input consisted of a burst of five stimuli at 200 Hz, reproducing the propensity of granule cells to burst at high frequencies (*Chadderton et al., 2004*; *Jörntell and Ekerot, 2006*). The climbing fibre input reflected the fact that these can occur in very high-frequency bursts (*Eccles et al., 1966*; *Maruta et al., 2007*). We used two stimuli at 400 Hz to represent the perturbation complex spike and four for the subsequent error complex spike if it was included in the protocol. Depending on the protocol, the climbing fibre stimuli had different timings relative to the granule cell stimuli: a pair of climbing fibre stimuli at 400 Hz, 11–15 ms or ∼500 ms after the start of the granule cell burst and/or four climbing fibre stimuli at 400 Hz, 100–115 ms after the beginning of the granule cell burst (timing diagrams will be shown in the Results). In a fraction of cells, the climbing fibre stimuli after the first were not reliable; our grounds for including these cells are detailed in the Materials and methods. The interval between the two bursts of climbing fibre stimuli when the error complex spike was present was about 100 ms. We increased the interval between induction episodes from the standard one second to two, to reduce any accumulating signal during induction. 300 such induction sequences were applied (*Lev-Ram et al., 2002*).

We first show the protocols relating to LTP (*Figure 4*). A granule cell burst was followed by a distant perturbation climbing fibre stimulus or the two inputs were activated simultaneously. In the examples shown, the protocol with simultaneous activation (GP_, *Figure 4C,D*) caused a potentiation of about 40%, while the temporally separate inputs caused a smaller change of 15% in the opposite direction (G_ _, *Figure 4A,B*). We note that individual outcomes were quite variable; group data and statistics will be shown below. The mean paired-pulse ratio in our recordings was A2/

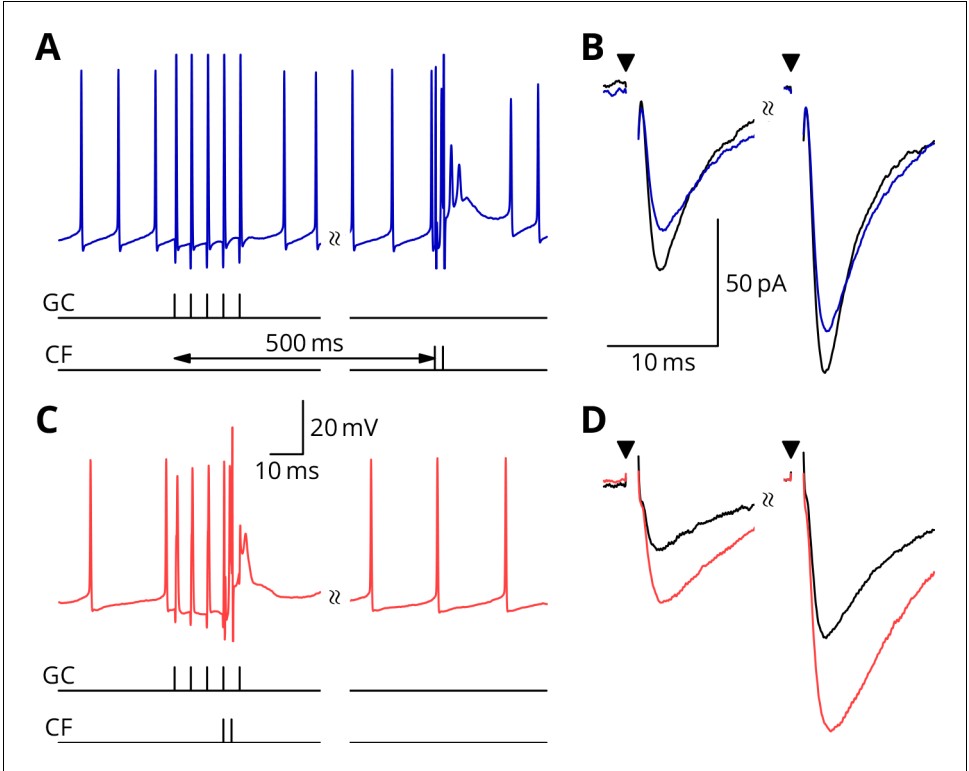

**Figure 4.** Simultaneous granule cell and climbing fibre activity induces LTP. (**A**) Membrane potential (*blue*) of a Purkinje cell during an induction protocol (G_ _) where a burst of 5 granule cell stimuli at 200 Hz was followed after 0.5 s by a pair of climbing fibre stimuli at 400 Hz. (**B**) Average EPSCs recorded up to 10 min before (*black*) and 20–30 min after the end of the protocol of A (*blue*). Paired test stimuli (*triangles*) were separated by 50 ms and revealed the facilitation typical of the granule cell input to Purkinje cells. In this case, the induction protocol resulted in a small reduction (*blue* vs. *black*) of the amplitude of responses to both pulses. (**C**) Purkinje cell membrane potential (*red*) during a protocol (GP_) where the granule cells and climbing fibres were activated simultaneously, with timing otherwise identical to A. (**D**) EPSCs recorded before (*black*) and after (*red*) the protocol in C. A clear potentiation was observed in both of the paired-pulse responses.
DOI: https://doi.org/10.7554/eLife.31599.005

A1 = 1.75 ± 0.32 (mean ± sd, $n$ = 58). As here, no significant differences of paired-pulse ratio were observed with any of the plasticity protocols: plasticity − baseline difference for GP_, mean −0.08, 95% confidence interval (−0.34, 0.20), $n$ = 15; GPE, mean 0.12, 95 % c.i. (−0.07, 0.33), $n$ = 10; G_ _, mean −0.01, 95 % c.i. (−0.24, 0.24), $n$ = 18; G_E, mean −0.09, 95 % c.i. (−0.23, 0.29), $n$ = 15.

*Figure 5* illustrates tests of our predictions regarding the induction of LTD. As before, a granule cell burst was paired with the perturbation climbing fibre, but now a longer burst of climbing fibre stimuli was appended 100 ms later, representing an error complex spike (GPE, *Figure 5C,D*). A clear LTD of about 40% developed following the induction. In contrast, if the perturbation complex spike was omitted, leaving the error complex spike (G_E, *Figure 5A,B*), no clear change of synaptic weight occurred (an increase of about 10%). During induction, cells would generally begin in a tonic firing mode, but nearly all ceased firing by the end of the protocol. The specimen sweeps in *Figure 5* are taken towards the end of the induction period and illustrate the Purkinje cell responses when spiking had ceased.

The time courses of the changes of EPSC amplitude are shown in normalised form in *Figure 6* (see Materials and methods). The individual data points of the relative EPSC amplitudes for the different protocols are also shown. A numerical summary of the group data and statistical comparisons is given in *Table 1*.

In a complementary series of experiments, we explored the plasticity outcome when six climbing fibre stimuli were grouped in a burst and applied simultaneously with the granule cell burst. This

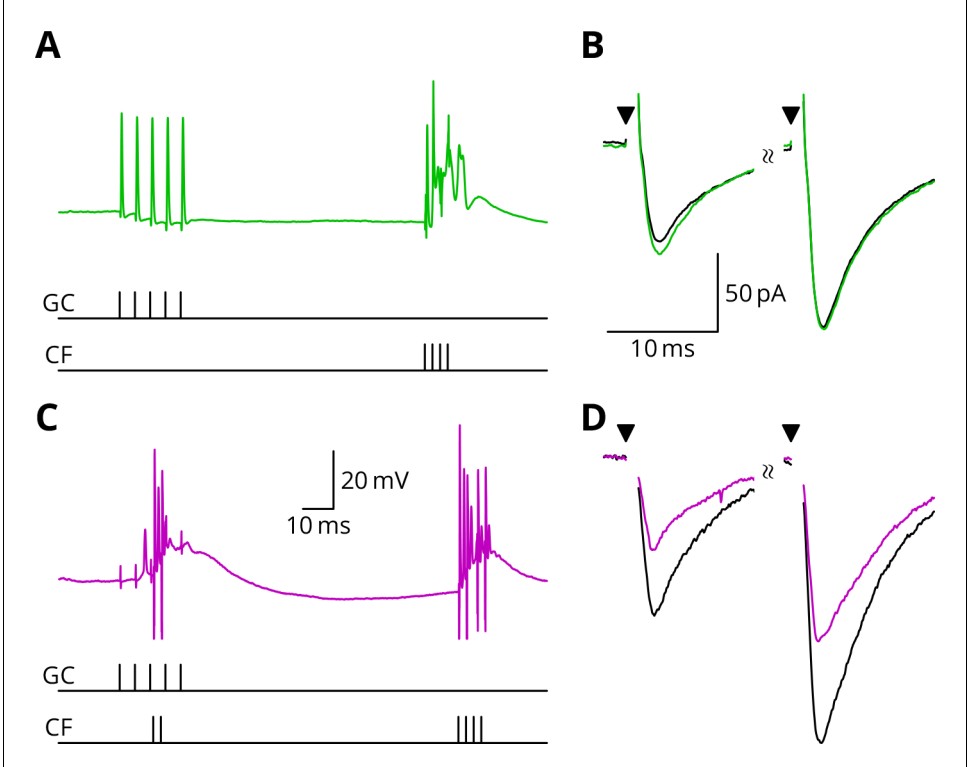

**Figure 5.** LTD requires simultaneous granule cell and climbing fibre activity closely followed by an additional complex spike. (**A**) Membrane potential of a Purkinje cell (*green*) during a protocol where a burst of five granule cell stimuli at 200 Hz was followed after 100 ms by four climbing fibre stimuli at 400 Hz (G_E). (**B**) Average EPSCs recorded up to 10 min before (*black*) and 20–30 min after the end of the protocol of A (*green*). The interval between the paired test stimuli (*triangles*) was 50 ms. The induction protocol resulted in little change (*green* vs. *black*) of the amplitude of either pulse. (**C**) Purkinje cell membrane potential (*magenta*) during the same protocol as in A with the addition of a pair of climbing fibre stimuli simultaneous with the granule cell stimuli (GPE). (**D**) EPSCs recorded before (*black*) and after (*magenta*) the protocol in C. A clear depression was observed.
DOI: https://doi.org/10.7554/eLife.31599.006

allowed comparison with the GPE protocol, which also contained 4 + 2 = 6 climbing fibre stimuli. The results in *Figure 6—figure supplement 1* shows that a modest LTP was observed on average: after/before ratio 1.12 ± 0.12 (mean ± SEM); 95% c.i. 0.89–1.35. This result is clearly different from the LTD observed under GPE ($p$=0.0034; two-tailed Wilcoxon rank sum test, after Bonferroni correction for four possible comparisons).

These results therefore provide experimental support for all four plasticity rules predicted by our proposed mechanism of stochastic gradient descent. We argue in the Discussion that the apparent contradiction of these results with the literature is not unexpected if the likely effects of our altered conditions are considered in the light of known mechanisms of potentiation and depression.

## Extraction of the change of error

Above we have provided experimental evidence in support of the counterintuitive synaptic plasticity rules predicted by our proposed learning mechanism. In that mechanism, following a perturbation complex spike, the sign of plasticity is determined by the absence or presence of a follow-up error complex spike that signals whether the movement error increased (spike present) or decreased (spike absent). We now return to the outstanding problem of finding a mechanism able to extract this *change* of error, $\delta\mathcal{E}$.

Several roughly equivalent schemes have been proposed (*Williams, 1992*), including subtraction of the average error (*Barto et al., 1983*) and decorrelation (*Dean and Porrill, 2014*), a specialisation of the covariance rule (*Sejnowski, 1977*). However, in general, these suggestions have not extended

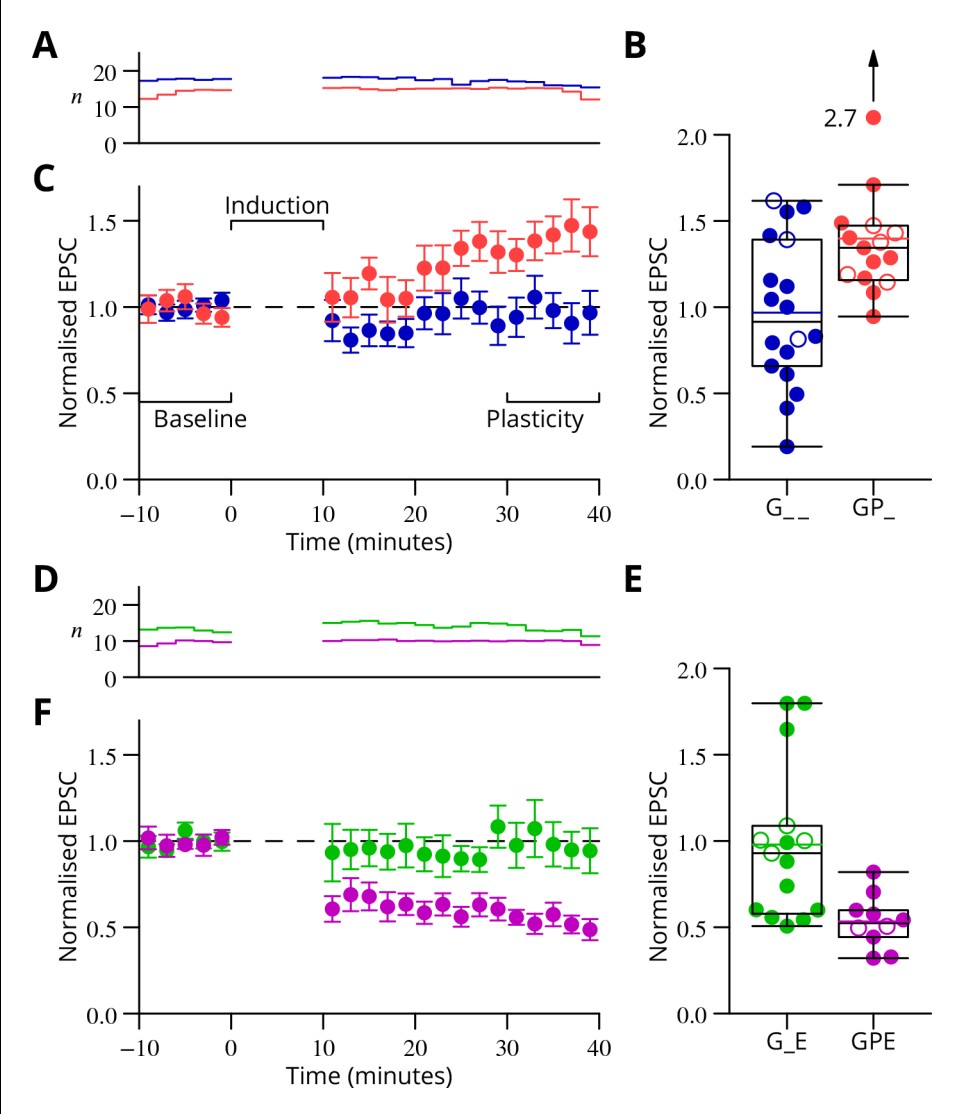

**Figure 6.** Time course and amplitude of plasticity. (**A**) number, (**B**) box-and-whisker plots of individual plasticity ratios (*coloured lines* represent the means, *open symbols* represent cells with failures of climbing fibre stimulation; see Materials and methods) and (**C**) time course of the mean EPSC amplitude for GP_ (red) and G_ _ (blue) protocols of *Figure 4*, normalised to the pre-induction amplitude. Averages every 2 min, mean ± sem. Non-integer *n* arise because the numbers of responses averaged were normalised by those expected in two minutes, but some responses were excluded (see Materials and methods) and some recordings did not extend to the extremities of the bins. Induction lasted for 10 min starting at time 0. (**D, E**) and (**F**) similar plots for the GPE (*magenta*) and G_E (*green*) protocols of *Figure 5*.

DOI: https://doi.org/10.7554/eLife.31599.007

The following figure supplement is available for figure 6:

**Figure supplement 1.** In complementary experiments, we examined the outcome of applying a burst of six climbing fibre stimuli simultaneously with the granule cell burst.

DOI: https://doi.org/10.7554/eLife.31599.008

to detailed cellular implementations. In order to restrict our implementation to biologically plausible mechanisms we selected a method that involves subtracting the average error from the trial-to-trial error (*Barto et al., 1983*; *Doya and Sejnowski, 1988*). The residual of the subtraction is then simply the variation of the error $\delta\mathcal{E}$ as desired.

**Table 1.** Group data and statistical tests for plasticity outcomes.
In the upper half of the table, the ratios of EPSC amplitudes after/before induction are described and compared with a null hypothesis of no change (ratio = 1). The GP_ and GPE protocols both induced changes, while the control protocols (G_ _, G_E) did not. The bottom half of the table analyses differences of those ratios between protocols. The 95% confidence intervals (c.i.) were calculated using bootstrap methods, while the $p$-values were calculated using a two-tailed Wilcoxon rank sum test. The p-values marked with an asterisk have been corrected for a two-stage analysis by a factor of 2 (Materials and methods).

| Comparison | Mean | 95 % c.i. | | $p$ | $n$ |
|---|---|---|---|---|---|
| GP_ | 1.40 | 1.26, | 1.72 | 0.0001 | 15 |
| GPE | 0.53 | 0.45, | 0.63 | 0.002 | 10 |
| G_ _ | 0.97 | 0.78, | 1.16 | 0.77 | 18 |
| G_E | 0.98 | 0.79, | 1.24 | 0.60 | 15 |
| GP_ vs G_ _ | 0.43 | 0.19, | 0.74 | 0.021* | |
| GPE vs G_E | −0.44 | −0.72, | −0.24 | 0.0018* | |
| GP_ vs G_E | 0.42 | 0.14, | 0.73 | 0.01* | |
| GPE vs G_ _ | −0.43 | −0.65, | −0.22 | 0.008* | |
| G_ _ vs G_E | 0.01 | −0.26, | 0.32 | 0.93 | |

DOI: https://doi.org/10.7554/eLife.31599.009

As mechanism for this subtraction, we propose that the excitatory synaptic drive to the inferior olive is on average balanced by input from the GABAergic nucleo-olivary neurones. A diagram is shown in *Figure 7* to illustrate how this might work in the context of repetitions of a single

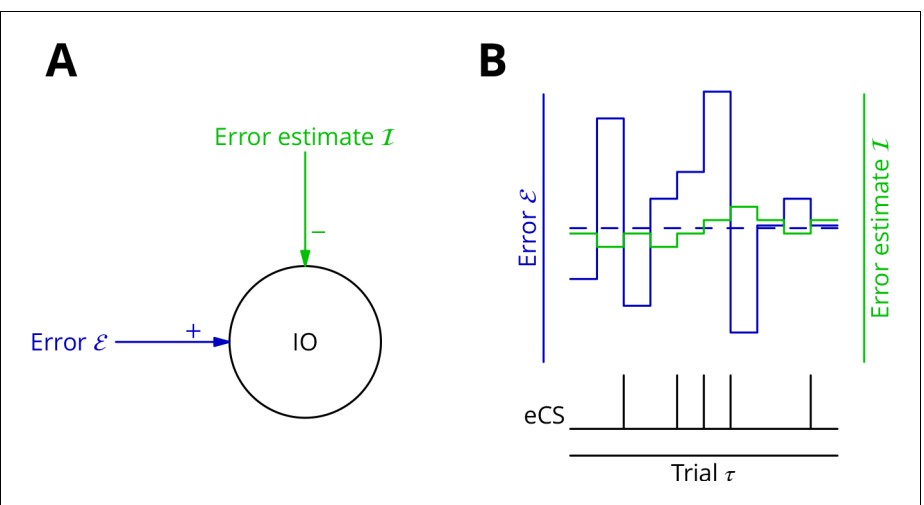

**Figure 7.** Adaptive tracking to cancel the mean error input to the inferior olive. (**A**) The olive is assumed to receive an excitatory signal representing movement error $\mathcal{E}$ and an inhibitory input $\mathcal{I}$ from the nucleo-olivary neurones of the cerebellar nuclei. (**B**) The inputs to the inferior olive are represented in discrete time (τ)—each bar can be taken to represent a discrete movement realisation. The error (*blue*) varies about its average (*dashed blue line*) because perturbation complex spikes influence the movement and associated error randomly. The strength of the inhibition is shown by the *green* trace. When the excitatory error input exceeds the inhibition, an error complex spike is emitted (*bottom black trace*) and the inhibition is strengthened by plasticity, either directly or indirectly. In the converse situation and in the consequent absence of an error complex spike, the inhibition is weakened. In this way, the inhibition tracks the average error and the emission of an error complex spike signals an error exceeding the estimated average. Note that spontaneous perturbation complex spikes are omitted from this diagram.
DOI: https://doi.org/10.7554/eLife.31599.010

movement (we extend the mechanism to multiple interleaved movements below). Briefly, a feedback plasticity reinforces the inhibition whenever it is too weak to prevent an error complex spike from being emitted. When the inhibition is strong enough to prevent an error complex spike, the inhibition is weakened. If the variations of the strength of the inhibition are sufficiently small and the error does not change rapidly, the level of inhibition will attain a good approximation of the average error. Indeed, this mechanism can be viewed as maintaining an estimate of the movement error. However, the error still varies about its mean on a trial-to-trial basis because of the random perturbations that influence the movement and therefore the error. In consequence, error complex spikes are emitted when the error exceeds the (estimated) average; this occurs when the perturbation increases the error. This mechanism enables extraction of the sign of $\delta\mathcal{E}$ in the context of a single movement realisation. In support of such a mechanism, there is evidence that inhibition in the olive builds up during learning and reduces the probability of complex spikes (*Kim et al., 1998*).

More than one plasticity mechanism could produce the desired cancellation of excitatory drive to the inferior olive. We outline two possibilities here, but it will be necessary in the implementation below to make a concrete if somewhat arbitrary choice; we shall make it on the basis of the available, circumstantial evidence.

The first possible mechanism would involve plasticity of the inhibitory synapses made by nucleo-olivary neurones in the inferior olive (*Figure 1A,B*). Perturbation and error complex spikes would be distinguished in an appropriate plasticity rule by the presence of excitatory synaptic input to the olive. This would offer a simple implementation, since plastic and cancelled inputs would be at neighbouring synapses (*De Zeeuw et al., 1998*); information about olivary spikes would also be directly available. However, the lack of published evidence and our own unsuccessful exploratory experiments led us to consider an alternative plasticity locus.

A second possible implementation for cancelling the average error signal would make the mossy fibre to nucleo-olivary neurone synapses plastic. The presence of an error complex spike would need to potentiate these inputs, thereby increasing inhibitory drive to the olive and tending to reduce the likelihood of future error complex spikes being emitted. Inversely, the absence of the error complex spike should depress the same synapses. Movement specificity could be conferred by applying the plasticity only to active mossy fibres, the patterns of which would differ between movements. This would enable movement-specific cancellation as long as the overlap between mossy fibre patterns was not too great.

How would information about the presence or absence of the error complex spike be supplied to the nucleo-olivary neurones? A direct connection between climbing fibre collaterals and nucleo-olivary neurones exists (*De Zeeuw et al., 1997*), but recordings of cerebellar neurones following stimulation of the olive suggest that this input is not strong, probably eliciting no more than a single spike per activation (*Bengtsson et al., 2011*). The function of this apparently weak input is unknown.

An alternative route to the cerebellar nuclear neurones for information about the error complex spike is via the Purkinje cells. Climbing fibres excite Purkinje cells which in turn inhibit cerebellar nuclear neurones, in which a strong inhibition can cause a distinctive rebound of firing (*Llinás and Mühlethaler, 1988*). It has been reported that peripheral stimulation of the climbing fibre receptive field, which might be expected to trigger the emission of error complex spikes, causes large IPSPs and an excitatory rebound in cerebellar nuclear neurones (*Bengtsson et al., 2011*). These synaptically induced climbing fibre–related inputs were stronger than spontaneously occurring IPSPs. In our conceptual framework, this could be interpreted as indicating that error complex spikes are stronger and/or arise in a greater number of olivary neurones than perturbation complex spikes. The two types of complex spike would therefore be distinguishable, at least in the cerebellar nuclei.

Plasticity of active mossy fibre inputs to cerebellar nuclear neurones has been reported which follows a rule similar to that our implementation requires. Thus, mossy fibres that burst before a hyperpolarisation (possibly the result of an error complex spike) that triggers a rebound have their inputs potentiated (*Pugh and Raman, 2008*), while mossy fibres that burst without a succeeding hyperpolarisation and rebound are depressed (*Zhang and Linden, 2006*). It should be noted, however, that this plasticity was studied at the input to projection neurones and not at that to the nucleo-olivary neurones. Nevertheless, the existence of highly suitable plasticity rules in a cell type closely related to the nucleo-olivary neurones encouraged us to choose the cerebellar nuclei as the site of the plasticity that leads to cancellation of the excitatory input to the olive.

We now consider how synaptic integration in the olive leads to emission or not of error complex spikes. The nucleo-olivary synapses (in most olivary nuclei) display a remarkable degree of delayed and long-lasting release (*Best and Regehr, 2009*), suggesting that inhibition would build up during a command and thus be able to oppose the excitatory inputs signalling movement errors that appear some time after the command is transmitted. The error complex spike would therefore be produced (or not) after the command. On this basis, we shall make the simplifying assumption that the cerebellum generates a relatively brief motor control output or 'command', of the order of 100 ms or less and a single error calculation is performed after the end of that command. As for the saccade example previously mentioned, many movements can only be evaluated after completion. In effect, this corresponds to an offline learning rule.

## Simulations

Above we outlined a mechanism for extracting the error change $\delta\mathcal{E}$; it is based on adapting the inhibitory input to the inferior olive to cancel the average excitatory error input in a movement-specific manner. To verify that this mechanism could operate successfully in conjunction with the scheme for cortical plasticity already described, we turned to simulation.

A reduced model of a cerebellar microzone was developed and is described in detail in the Materials and methods. In overview, mossy fibre input patterns drove Purkinje and cerebellar nuclear neurones during commands composed of 10 discrete time bins. Purkinje cell activity was perturbed by randomly occurring complex spikes, which each increased the firing in a single time bin. The learning task was to adjust the output patterns of the nuclear projection neurones to randomly chosen targets. Cancellation of the average error was implemented by plasticity at the mossy fibre to nucleo-olivary neurone synapse while modifications of the mossy fibre pathway input to Purkinje cells reflected the rules for stochastic gradient descent described earlier. Synaptic weights were updated offline after each command. The global error was the sum of absolute differences between the projection neurone outputs and their target values. Error complex spikes were broadcast to all Purkinje cells when the error exceeded the integral of inhibitory input to the olive during the movement. There were thus 400 (40 projection neurones $\times$ 10 time bins) variables to optimise using a single error value.

The progress of the simulation is illustrated in *Figure 8*, in which two different movements were successfully optimised in parallel; only one is shown. The error can be seen to decrease throughout the simulation, indicating the progressive improvement of the learnt command. The effect of learning on the difference between the output and the target values can be seen in *Figure 8C and D*. The initial match is very poor because of random initial weight assignments, but it has become almost exact by the end of the simulation.

The optimisation only proceeds once the inhibitory input to the olive has accurately subtracted the average error. Thus, in *Figure 8B* it can be seen that the initial values of the inhibitory and excitatory (error) inputs to the olive differed. The inhibition tends towards the error. Until the two match, the overall error shows no systematic improvement. This confirms the need for accurate subtraction of the mean error to enable extraction of the error changes necessary to descend the error gradient. This simulation supports the feasibility of our proposed cerebellar implementation of stochastic gradient descent.

## Algorithm convergence and capacity

The simulations above provide a proof of concept for the proposed mechanism of cerebellar learning. Nonetheless, even for the relatively simple network model in the simulations, it is by no means obvious to determine the regions of parameter space in which the model converges to the desired outputs. It is also difficult to analyse the algorithm's performance compared to other more classical ones. To address these issues, we abstract the core mechanism of stochastic gradient descent with estimated global errors in order to understand better the algorithm dynamics and to highlight the role of four key parameters. Analysis of this mechanism shows that this algorithm, even in this very reduced form, exhibits a variety of dynamical regimes, which we characterise. We then show how the key parameters and the different dynamical learning regimes directly appear in an analog perceptron description of the type considered in the previous section. We find that the algorithm's storage capacity is similar to the optimal capacity of an analog perceptron.

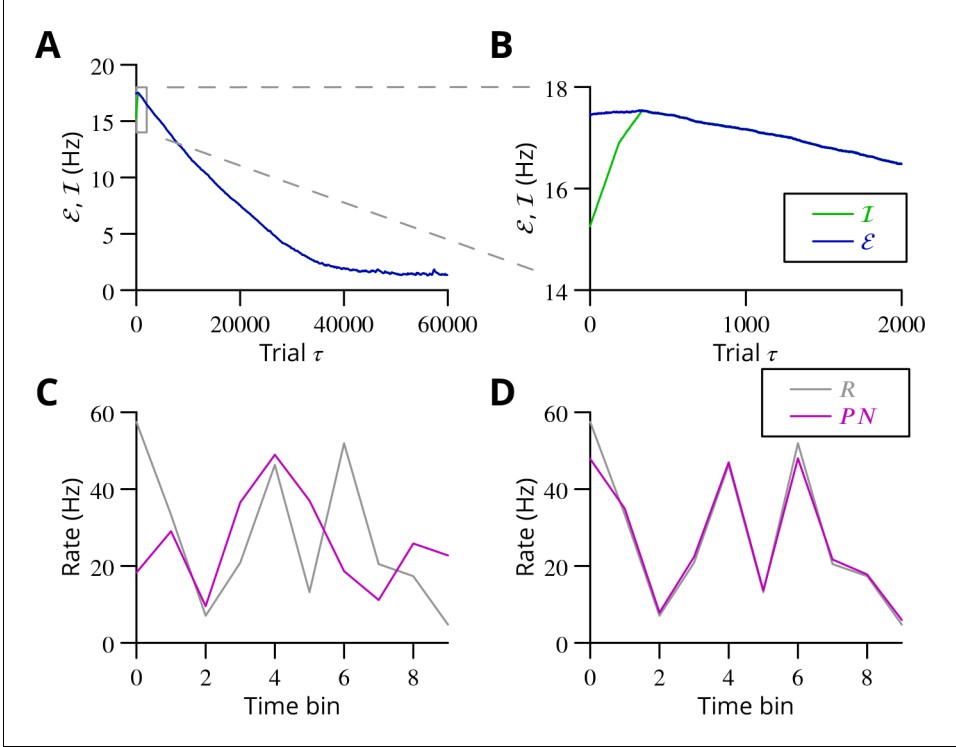

**Figure 8.** Simulated cerebellar learning by stochastic gradient descent with estimated global errors. The total error ($\mathcal{E}$, *blue*) at the cerebellar nuclear output and the cancelling inhibition ($\mathcal{I}$, *green*) reaching the inferior olive are plotted as a function of trial number (τ) in (**A** and **B**) for one of two interleaved patterns learnt in parallel. An approximately 10-fold reduction of error was obtained. It can be seen in A that the cancelling inhibition follows the error very closely over most of the learning time course. However, the zoom in B shows that there is no systematic reduction in error until the inhibition accurately cancels the mean error. (**C**) Initial firing profile of a typical cerebellar nuclear projection neurone (*PN*, *magenta*); the simulation represented 10 time bins with initially random frequency values per neurone, with a mean of 30 Hz. The target firing profile for the same neurone (*R*, *grey*) is also plotted. (**D**) At the end of the simulation, the firing profile closely matched the target.
DOI: https://doi.org/10.7554/eLife.31599.011

## Reduced model

In order to explore more exhaustively the convergence of the learning algorithm, we considered a reduced version with a single principal cell that nevertheless captures its essence (*Figure 9A*). A detailed analysis of this circuit is presented in the Appendix 1, which we summarise and illustrate here.

We focus on the rate $P$ of a single P-cell (principal cell, in essence a combination of a Purkinje cell and a nuclear projection neurone), without considering how the firing is driven by synaptic inputs. The other variable of the model is the strength $\mathcal{J}$ of the plastic excitatory input from mossy fibres to nucleo-olivary neurones *Figure 9*. The learning task is to bring the firing rate $P$ of the P-cell to a desired target rate $R$, guided by an estimation of the current error relying on $\mathcal{J}$, which is refined concurrently with $P$ from trial to trial.

The error $\mathcal{E}$ is determined by the difference between $P$ and $R$; we choose the absolute difference $\mathcal{E} = |P - R|$. In the presence of a perturbation $A$, occurring with probability $\rho$, the rate becomes $P + A$ and so $\mathcal{E} = |P + A - R|$. The current estimate of the error is measured by the strength $\mathcal{I}$ of the inhibition on the olivary neurone associated with the P-cell. It assumes the value $\mathcal{I} = [\mathcal{J} - qP]_+$ (the brackets $[X]_+$ denoting the rectified value of X, $[X]_+ = X$ if $X \geq 0$ and 0 otherwise; $q$ represents the strength of P-cell (in reality Purkinje cell) synapses on nucleo-olivary neurones. The inhibition of the inferior olive (IO) arises from the discharge of the nucleo-olivary neurones (NO) induced by the mossy fibre inputs, the $\mathcal{J}$ component of $\mathcal{I}$. The NOs are themselves inhibited by the P-cell which is accounted for by the $-qP$ component of $\mathcal{I}$ (*Figure 9A*). Note that in the presence of a perturbation

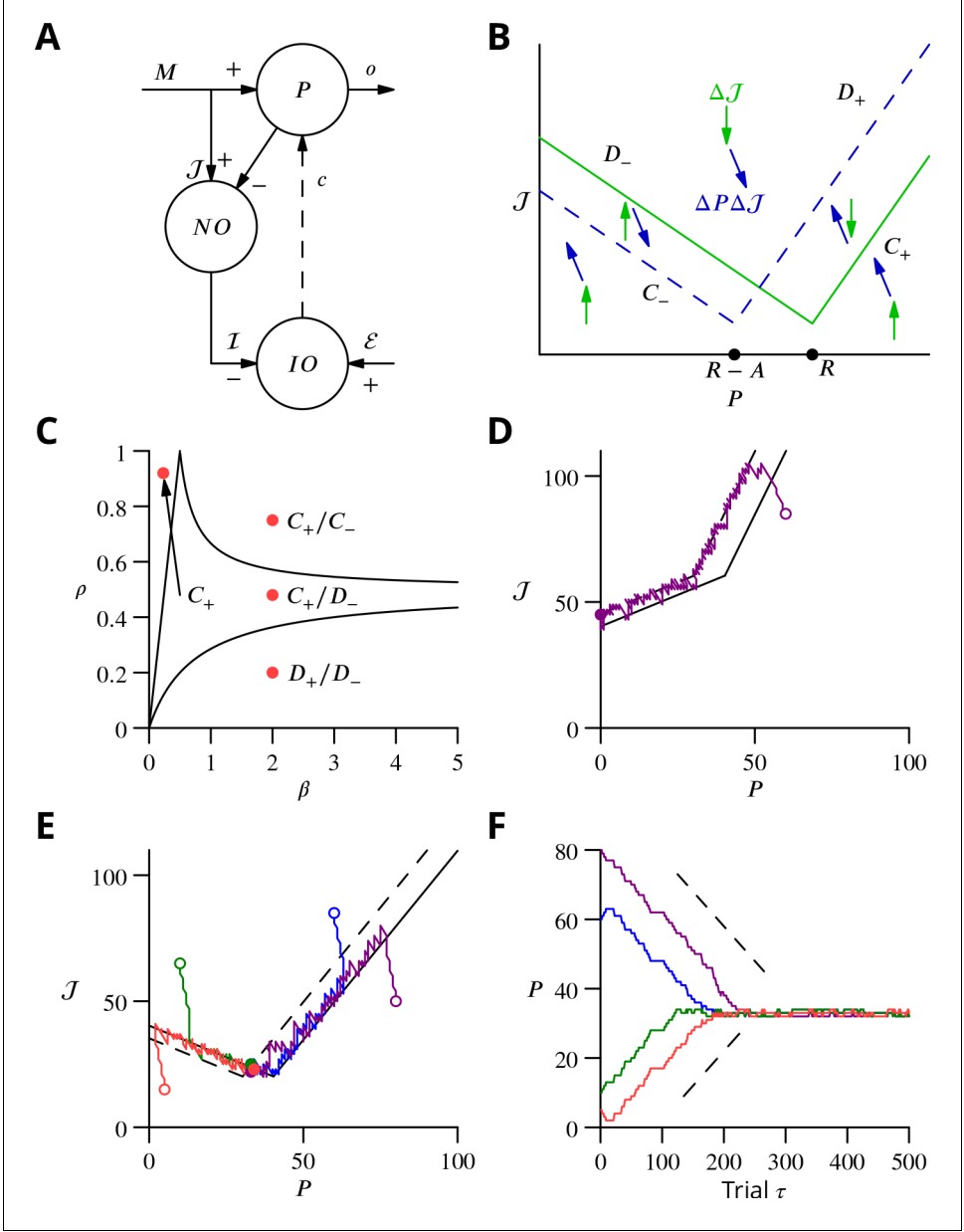

**Figure 9.** Single-cell convergence dynamics in a reduced version of the algorithm. (A) Simplified circuitry implementing stochastic gradient descent with estimated global errors. Combined Purkinje cells/projection neurones (P) provide the output $o$. These cells receive an excitatory plastic input from mossy fibres (M). Mossy fibres also convey a plastic excitatory input $\mathcal{J}$ to the nucleo-olivary neurones (NO), which receive inhibitory inputs from the Purkinje cells and supply the inhibitory input $\mathcal{I} = [\mathcal{J} - qP]_+$ to the inferior olive (IO). The inferior olive receives an excitatory error input $\mathcal{E}$. The olivary neurones emit spikes that are transmitted to the P-cells via the climbing fibre $c$. (B) Effects of plasticity on the simplified system in the plane defined by the P-cell rate $P$ (with optimum $R$, perturbation $A$) and excitatory drive to the nucleo-olivary neurones $\mathcal{J}$. Plastic updates of type $\Delta P \Delta \mathcal{J}$ (*blue arrows*) and $\Delta \mathcal{J}$ (*green arrows*) are shown. The updates change sign on the line C (*dashed blue*) and D (*solid green*), respectively. Lines $C_\pm$ and $D_\pm$ delimit the 'convergence corridors'. The diagram is drawn for the case $q<1$, in which $A$, the perturbation of $P$ is larger than the perturbation $qA$ of $\mathcal{I}$. (C) Parameters $\beta$ and $\rho$ determine along which lines the system converges to $R$. The *red dot* in the $D_+/D_-$ region shows the parameter values used in panels E and F and also corresponds to the effective parameters of the perceptron simulation of *Figure 10*. The other *red dots* show the parameters used in the learning examples displayed in *Appendix 1—figure 1*. (D) When $q>1$, $C_+$ and $D_-$ do not cross and learning does not converge to the desired rate. After going down the $C_+/D_+$ corridor, the point $(P, \mathcal{J})$ continues along the $C_-/D_-$ corridor without stopping close to the desired target rate $R$. *Figure 9 continued on next page*

*Figure 9 continued*

*Open circle*, start point; *filled circle*, end point. (**E**) Dynamics in the $(P, \mathcal{J})$ plane for $\rho = 0.2$ and $\beta = 2$
$(A = 10, \Delta P = 1, \Delta \mathcal{J} = 2, q = 0.5)$. Trajectories (*coloured lines*) start from different initial conditions (*open circles*).
*Open circles*, start points; *filled circles*, end points. (**F**) Time courses of $P$ as a function of trial $\tau$ for the trajectories
in C (*same colours*). *Dashed lines*: predicted rate of convergence (Appendix 1). All trajectories end up fluctuating
around $P = R - A(q + 1)/2$, which is $R - 7.5$ with the chosen parameters. *Open circles*, start points; *filled circles*,
end points.
DOI: https://doi.org/10.7554/eLife.31599.012

of the P-cell, the estimate of the error is itself modified to $[\mathcal{J} - qP - qA]_+$ to account for the decrease (for $q>0$) of the NO discharge arising from the P-cell firing rate increase.

The system learning dynamics can be analysed in the $P$-$\mathcal{J}$ plane (*Figure 9B*). After each 'movement' realisation, the two system variables, $P$ and $\mathcal{J}$ are displaced within the $P$-$\mathcal{J}$ plane by plasticity according to the algorithm. In the presence of a perturbation, $P$ is decreased (a leftwards displacement) if $\mathcal{E}>[\mathcal{J} - qP - qA]_+$ and otherwise increased. However, $P$ remains unchanged if there is no perturbation. Conversely, $\mathcal{J}$ is always updated irrespective of the presence or absence of a perturbation. If $\mathcal{E}>[\mathcal{J} - qP - qA]_+$ in the presence of a perturbation or $\mathcal{E}>\mathcal{J} - qP$ in its absence, $\mathcal{J}$ is increased (upwards displacement) and it is decreased otherwise.

Updates can therefore be described as $\Delta P \Delta \mathcal{J}$ or $\Delta \mathcal{J}$. The resultant directions vary between the different regions of the $(P, \mathcal{J})$-plane (*Figure 9B*) delimited by the two borders $C$ and $D$, defined as $\mathcal{J} - qP - qA = |P - R + A|$ and $\mathcal{J} - qP = |P - R|$, respectively. The two half-lines bordering each sector are denoted by a plus or minus index according to their slopes.

Stochastic gradient descent is conveniently analysed with a phase-plane description, by following the values $(P, \mathcal{J})$ from one update to the next. The dynamics randomly alternate between updates of type $\Delta P \Delta \mathcal{J}$ and $\Delta \mathcal{J}$ which leads $(P, \mathcal{J})$ to follow a biased random walk in the $(P, \mathcal{J})$ plane. Mathematical analysis (see Appendix 1) shows that the dynamics proceed in three successive phases, as observed in the simulations of *Figure 8*. First, the pair of values $(P, \mathcal{J})$ drifts from the initial condition towards one of the two 'corridors' between the lines $C_+$ and $D_+$ or between $C_-$ and $D_-$ (see *Figure 9B*). This first phase leads the estimated error $\mathcal{I}$ to approximate closely the error $\mathcal{E}$, as seen in the full network simulation (*Figure 8*).

When a corridor is reached, in a second phase, $(P, \mathcal{J})$ follows a stochastic walk in the corridor with, under suitable conditions, a mean linear decrease of the error in time with bounded fluctuations. The precise line followed and the mean rate of error decrease depends on the initial conditions and two parameters: $\rho$ the probability of a perturbation occurring and $\beta = \Delta \mathcal{J}/\Delta P$, as indicated in *Figure 9C*. Typical trajectories and time courses for specific values of $\rho$ and $\beta$ and different initial conditions are shown in *Figure 9D,E,F* as well as in *Appendix 1—figure 1*.

Error decrease in this second phase requires certain restrictions upon the four key parameters—$\rho$, $\beta$, $\Delta P$, $\Delta \mathcal{J}$. These restrictions are that $0<\rho<1$, which ensures that updates are not restricted to a line, and $\Delta \mathcal{J}<(1 - q)A$ as well as $\Delta P + \Delta \mathcal{J}/(1 + q)<A$, which are sufficient to ensure that a trajectory will eventually enter a convergence corridor (large updates might always jump over the corridor).

In a final phase, $(P, \mathcal{J})$ fluctuates around the intersection of $C_+$ and $D_-$, when it exists.

$C_+$ and $D_-$ do not cross for $q>1$, when the perturbation $qA$ of the estimated error $\mathcal{I}$ is larger than $A$ the perturbation of the P-cell discharge. In this case, $P$ does not stop close to its target value, as illustrated in *Figure 9D*. For $q<1$, $C_+$ and $D_-$ intersect at $(R - A(q + 1)/2, qR + (1 - q^2)A/2)$. The final error in the discharge rate fluctuates around $A(q + 1)/2$. Namely, the mean final error grows from $A/2$ when $q = 0$ (vanishing inhibition of the nucleo-olivary discharge by the P-cell) to $A$ when $q = 1$ (maximal admissible inhibition of the nucleo-olivary discharge by the P-cell). The failed convergence for $q \geq 1$ could be expected, since in this scenario the perturbation corrupts the error estimate more than it influences the global error due to the perturbed P-cell firing rate.

## Analog perceptron

In order to extend the above analysis to include multiple synaptic inputs and to explore the algorithm's capacity to learn multiple movements, we considered an analog perceptron using the algorithm of stochastic gradient descent with estimated global errors. This allowed us to investigate the

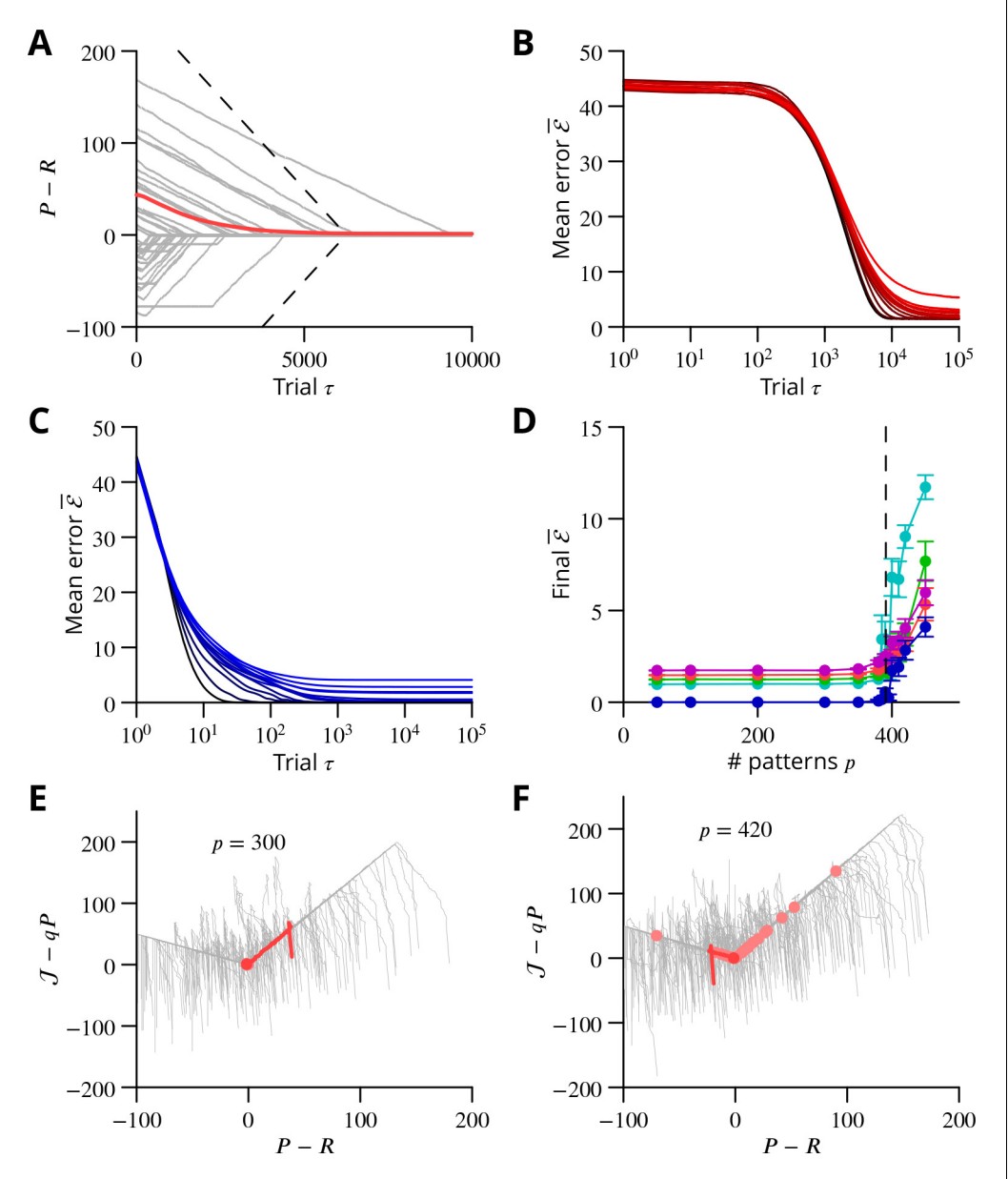

**Figure 10.** Convergence and capacity for an analog perceptron. (A) Convergence for a single learned pattern for stochastic gradient descent with estimated global errors (SGDEGE). Different lines (*thin grey lines*) correspond to distinct simulations. The linear decrease of the error $(P - R)$ predicted from the simplified model without zero-weight synapses is also shown (*dashed lines*). When learning increases the rate towards its target, the predicted convergence agrees well with the simulation. When learning decreases the rate towards its target, the predicted convergence is larger than observed because a fraction of synapses have zero weight. (B) Mean error vs. number of trials (always per pattern) for different numbers of patterns for SGDEGE with $q = 0.5$. *Colours from black to red*: 50, 100, 200, 300, 350, 380, 385, 390, 395, 400, 410, 420, 450. (C) Mean error vs. number of trials when learning using the delta rule. The final error is zero when the number of patterns is below capacity (predicted to be 391 for these parameters), up to finite size effects. *Colours from black to blue*: same numbers of patterns as listed for B. (D) Mean error after $1 \times 10^5$ trials for the delta rule (*blue*) and for the SGDEGE with $q = 0$ (*cyan*), 0.25 (*green*), 0.5 (*red*), 0.75 (*magenta*), as a function of the number of patterns $p$. The mean error diverges from its minimal value close to the theoretical capacity (*dashed line*) for both learning algorithms. (E) Dynamics of pattern learning in the $P - \mathcal{J}$ or more precisely $(P - R) - (\mathcal{J} - qP)$ plane below maximal capacity ($p = 300$). The error $(P - R)$ corresponding to each pattern (*grey*) is reduced as it moves down its convergence corridor. The trajectory for a single specimen pattern is highlighted in *red*, while the endpoints of all trajectories are indicated by *salmon filled*

*Figure 10 continued on next page*

*Figure 10 continued*

*circles* but are all obscured at (0,0) by the *red filled circle* of the specimen trajectory. (F) Same as in E for a number of patterns above maximal capacity ($p = 420$). After learning several patterns, rates remain far from their targets (*salmon filled circles*). The SGDEGE algorithm parameters used to generate this figure are $A = 2, \Delta \mathcal{J} = 0.4, \Delta P = 0.2, \rho = 0.2$. The parameter $q = 0.5$ except in panel D where it is explicitly varied. The analog perceptron parameters are $P_{max} = 100, f = 0.2, N_m = 1000$ and the threshold $\theta = 12.85$ with $\gamma = 1$.

DOI: https://doi.org/10.7554/eLife.31599.013

storage capacity attainable with the algorithm and to compare it to the theoretical maximal capacity for an analog perceptron (*Clopath and Brunel, 2013*).

The architecture was again that of *Figure 9A*; the details of the methods can be found in Appendix 1 and we summarise the conclusions here, with reference to *Figure 10*. The simulation included 1000 inputs (mossy fibres) to a P-cell; note that much larger numbers of inputs pertain in vivo (*Napper and Harvey, 1988*).

As found in the reduced model above, convergence of the algorithm requires the perturbation of the error estimate to be smaller than the effect of the P-cell perturbation on the true error, namely $q<1$. When this condition holds, the rate of learning a single pattern and the final error can be related to that predicted from the simplified analysis above (*Figure 10A* and in Appendix 1).

Learning using perturbations is slower than when using the full error information (i.e., the 'delta rule' (*Dayan and Abbott, 2001*); compare *Figure 10B and C*), for all numbers of patterns. The difference can be attributed to the different knowledge requirement of the two algorithms. When the error magnitude is known, which requires knowledge of the desired endpoint, large updates can be made for large errors, as done in the delta rule. In contrast, the proposed SGDEGE algorithm does not require knowledge of the error absolute magnitude or sign and thus proceeds with constant updating steps. Extensions of the SGDEGE algorithm that incorporate adaptive weight updates and could accelerate learning are mentioned in the Discussion.

The precision of learning is limited by the on-going perturbations to $A(1 + q)/2$ (e.g. $A/2$ for $q = 0$ or, $1.5A$ for $q = 0.5$ as used in *Figure 10B*), but the final average error increases beyond this floor only very close to the theoretical capacity computed for the input-output association statistics (*Figure 10D*). The contribution of interfering patterns to the slowed learning is similar for the two algorithms. The behaviour of individual learning trajectories below maximal capacity is shown in *Figure 10E and F*.

These analyses establish the convergence mechanisms of stochastic gradient descent with estimated global errors and show that it can attain the maximum theoretical storage capacity up to a non-zero final error resulting from the perturbation. Learning is slower than when the full error information is used in the delta rule, but we argue that the availability of that information would not be biologically plausible for most complex movements.

## Discussion

### A cellular implementation of stochastic gradient descent

Analysis of the requirements and constraints for a general cerebellar learning algorithm highlighted the fact that the current consensus Marr-Albus-Ito model is only capable of learning simple reflex movements. Optimisation of complex, arbitrary movements, of which organisms are certainly capable and to which the cerebellum is widely believed to contribute, would require a different algorithm. We therefore sought to identify within the cerebellar system an implementation of stochastic gradient descent. This should comprise several elements: a source of perturbations, a mechanism for extracting the change of error, and a plasticity rule incorporating this information. We identified a strong constraint on any implementation, requiring each calculation to be made in the context of a single movement realisation. This arises from the potentially arbitrary sequencing of movements with different optima. We also sought a mechanism that only makes use of plausible cellular calculations: summation of excitation and inhibition in the presence of a threshold.

We suggest that the perturbation is provided by the complex spike, which has suitable properties: spontaneous irregular activity, an unambiguous sign during the action potential burst, salience

at a cellular and network level, and the ability to influence synaptic plasticity. This choice of perturbation largely determines the predicted cerebellar cortical plasticity rules: only granule cell inputs active at the same time as a perturbation complex spike undergo plasticity, whose sign is determined by the absence (LTP) or presence (LTD) of a succeeding error complex spike. We have provided evidence that the synaptic plasticity rules do operate as predicted, in vitro under conditions designed to be more physiological than is customary.

An additional plasticity mechanism seems to be required to read off the change of error. The general mechanism we propose involves subtraction of the average error to expose the random variations caused by the perturbations of the movement. The subtraction results from adaptive tracking of the excitatory input to the olivary neurones by the inhibitory input from the nucleo-olivary neurones of the cerebellar nuclei. We chose to place the plasticity at the mossy fibre–nucleo-olivary neurone synapse, mostly because of the existence of suitable plasticity rules at the mossy fibre synapse onto the neighbouring projection neurones. However, plasticity in the olive at the nucleo-olivary input would probably be functionally equivalent and we do not intend to rule out this or alternative sites of the error-cancelling plasticity.

By simulating a simplified cerebellar network implementing this mechanism, we established the ability of our proposed mechanism to learn multiple arbitrary outputs, optimising 400 variables per movement with a single error value. More formal analysis of a simplified version of stochastic gradient descent with estimated global errors established convergence of the algorithm and allowed us to estimate its storage capacity.

## Implications for studies of synaptic plasticity

The plasticity rules for parallel fibre–Purkinje cell synapses predicted by our algorithm appeared to be incompatible with the well-established consensus. However, we show that under different, arguably more physiological, conditions, we were able to provide support for the four predicted outcomes.

We made several changes to the experimental conditions, only one of which is specific to the cerebellum. Thus, keeping synaptic inhibition intact has long been recognised as being of potential importance, with debates regarding its role in hippocampal LTP dating back decades (*Wigström and Gustafsson, 1983a*; *Wigström and Gustafsson, 1983b*; *Arima-Yoshida et al., 2011*). Very recent work also highlights the importance of inhibition in the induction of cerebellar plasticity (*Rowan et al., 2018*; *Suvrathan and Raymond, 2018*).

We also made use of a lower extracellular calcium concentration than those almost universally employed in studies of plasticity in vitro. In vivo measurements of the extracellular calcium concentration suggest that it does not exceed 1.5 mM in rodents, yet most studies use at least 2 mM. A 25% alteration of calcium concentration could plausibly change plasticity outcomes, given the numerous nonlinear calcium-dependent processes involved in synaptic transmission and plasticity (*Nevian and Sakmann, 2006*; *Graupner and Brunel, 2007*).

A major change of conditions we effected was cerebellum-specific. Nearly all studies of granule cell–Purkinje cell plasticity have employed stimulation of parallel fibres in the molecular layer. Such concentrated, synchronised input activity is unlikely to arise physiologically. Instead of this, we stimulated in the granule cell layer, a procedure expected to generate a more spatially dispersed input on the Purkinje cell, presumably leading to minimised dendritic depolarisations. Changing the stimulation method has been reported to prevent induction of LTD using standard protocols (*Marcaggi and Attwell, 2007*).

Although we cannot predict in detail the mechanistic alterations resulting from these changes of conditions, it is nevertheless likely that intracellular calcium concentrations during induction will be reduced, and most of the changes we observed can be interpreted in this light. It has long been suggested that high calcium concentrations during induction lead to LTD, while lower calcium concentrations generate LTP (*Coesmans et al., 2004*); we have recently modelled the induction of this plasticity, incorporating both calcium and nitric oxide signalling (*Bouvier et al., 2016*). Consistently with this viewpoint, a protocol that under standard conditions produce LTD—simultaneous activation of granule cells and climbing fibres—could plausibly produce LTP in the present conditions as a result of reduced intracellular calcium. Analogously, granule cell stimulation that alone produces LTP under standard conditions might elicit no change if calcium signalling were attenuated under our conditions.

Interestingly, LTP resulting from conjunctive granule cell and climbing fibre stimulation has been previously reported, in vitro (*Mathy et al., 2009*; *Suvrathan et al., 2016*) and in vivo (*Wetmore et al., 2014*). In contrast, our results do not fit well with several other studies of plasticity in vivo (*Ito et al., 1982*; *Jörntell and Ekerot, 2002*; *Jörntell and Ekerot, 2003*; *Jörntell and Ekerot, 2011*). However, in these studies quite intense stimulation of parallel and/or climbing fibre inputs was used, which may result in greater depolarisations and calcium entry than usually encountered. This difference could therefore account for the apparent discrepancy with the results we predict and have found in vitro.

It is unlikely that the interval between perturbation and error complex spikes would be fixed from trial to trial and it is certainly expected to vary with sensory modality (vision being slow). Our theoretical framework would therefore predict that a relatively wide range of intervals should be effective in inducing LTD through the GPE protocol. However, this prediction is untested and it remains possible that different intervals (potentially in different cerebellar regions) may lead to different plasticity outcomes, as suggested by the recent work of *Suvrathan et al. (2016)*; this might also contribute to some of the variability of individual plasticity outcomes we observe.

Another open question is whether different relative timings of parallel and climbing fibre activity would result in different plasticity outcomes. In particular, one might hypothesise that parallel fibres active during a pause following a perturbation complex spike might display plasticity of the opposite sign to that reported here for synchrony with the complex spike itself.

In summary, while in vitro studies of plasticity are likely to reveal molecular mechanisms leading to potentiation and depression, the outcomes from given stimulation protocols may be very sensitive to the precise conditions, making it difficult to extrapolate to the in vivo setting, as we have shown here for the cerebellum. Similar arguments could apply to in vitro plasticity studies in other brain regions.

## Current evidence regarding stochastic gradient descent

As mentioned in the introductory sections, the general cerebellar learning algorithm we propose here is not necessarily required in situations where movements are simple or constrained, admitting a fixed mapping between errors and corrective action. Furthermore, such movements constitute the near totality of well-studied models of cerebellar learning. Thus, the vestibulo-ocular reflex and saccade adaptation involve eye movements, which are naturally constrained, while the eyeblink is a stereotyped protective reflex. There is therefore a possibility that our mechanism does not operate in the cerebellar regions involved in oculomotor behaviour, even if it does operate elsewhere.

In addition, these ocular behaviours apparently display error functions that are incompatible with our assumptions. In particular, disturbance of a well optimised movement would be expected to increase error. However, it has been reported multiple times that climbing fibre activity can provide directional error information, including *reductions* of climbing fibre activity below baseline (e.g. *Soetedjo et al., 2008*). This argument is not totally conclusive, however. Firstly, we recall that the error is represented by the input to the inferior olive, not its output. It is thus possible that inputs from the nucleo-olivary neurones (or external inhibitory inputs) to the olive also have their activity modified by the disturbance of the movement, causing the reduction of climbing fibre activity. Secondly, what matters for our algorithm is the temporal sequence of perturbation and error complex spikes, but investigation of these second-order statistics of complex spike activity in relation to plasticity has, to our knowledge, not been reported. Similarly, it has been reported that learning and plasticity (LTD) occur in the absence of *modulation* of climbing fibre activity (*Ke et al., 2009*). Although this is difficult to reconcile with either the standard theory or our algorithm, it does not entirely rule out the existence of perturbation-error complex spike pairs that we predict lead to LTD.

Trial-to-trial plasticity correlated with the recent history of complex spikes has been demonstrated in oculomotor adaptation (*Medina and Lisberger, 2008*; *Yang and Lisberger, 2014*). This suggests that one way of testing whether our algorithm operates would be to examine whether the history of complex spike activity can predict future changes in the simple spike firing rate, according to the plasticity rules described above. For instance, two complex spikes occurring at a short interval should cause at the time of the first an increase of simple spike firing in subsequent trials. However, the most complete datasets reported to date involve oculomotor control (*Yang and Lisberger, 2014*; *Catz et al., 2005*; *Soetedjo et al., 2008*; *Ke et al., 2009*), where, as mentioned above, our algorithm may not be necessary.

Beyond the predictions for the plasticity rules at parallel fibre–Purkinje cell synapses tested above, there are a number of aspects of our theory that do fit well with existing observations. The simple existence of spontaneous climbing fibre activity is one. Additional suggestive features concern the evolution of climbing fibre activity during eyeblink conditioning (*Ohmae and Medina, 2015*). Once conditioning has commenced, the probability of complex spikes in response to the unconditioned stimulus decreases, which would be consistent with the build-up of the inhibition cancelling the average error signal in the olive. Furthermore, omission of the unconditioned stimulus then causes a reduction in the probability of complex spikes below the baseline rate, strongly suggesting a specifically timed inhibitory signal has indeed developed at the time of the unconditioned stimulus (*Kim et al., 1998*).

We suggest that the cancellation of average error involves plasticity at mossy fibre–nucleo-olivary neurone synapses. To date no study has reported such plasticity, but the nucleo-olivary neurones have only rarely been studied. Plasticity at the mossy fibre synapses on projection neurones has been studied both in vitro (*Pugh and Raman, 2006*; *Pugh and Raman, 2008*; *Zhang and Linden, 2006*) and in vivo (*Ohyama et al., 2006*), but is not used in our proposed algorithm. Axonal remodelling and synaptogenesis of mossy fibres in the cerebellar nuclei may underlie this plasticity (*Kleim et al., 2002*; *Boele et al., 2013*; *Lee et al., 2015*) and could also contribute to the putative plasticity at mossy fibre synapses on nucleo-olivary neurones.

Finally, our theory of course predicts that perturbation complex spikes perturb ongoing movements. It is well established that climbing fibre activation can elicit movements (*Barmack and Hess, 1980*; *Kimpo et al., 2014*; *Zucca et al., 2016*), but it remains to be determined whether the movements triggered by spontaneous climbing fibre activity are perceptible. *Stone and Lisberger (1986)* reported the absence of complex-spike-triggered eye movements in the context of the vestibulo-ocular reflex. However, it is known that the visual system is very sensitive to retinal slip (*Murakami, 2004*), so it may be necessary to carry out high-resolution measurements and careful averaging to confirm or exclude the existence of perceptible movement perturbations.

## Climbing fibre receptive fields and the bicycle problem

If the cerebellum is to contribute to the optimisation of complex movements, its output controlling any given muscle must be adjustable by learning involving multiple error signals. Thus, it may be useful to adjust an arm movement using vestibular error information as part of a balancing movement, but using touch information in catching a ball. However, it is currently unclear to what extent individual Purkinje cells can access different error signals.

There is an extensive literature characterising the modalities and receptive fields of climbing fibres. The great majority of reports are consistent with climbing fibres having fixed, specific modalities or very restricted receptive fields, with neighbouring fibres having similar properties (*Garwicz et al., 1998*; *Jörntell et al., 1996*). Examples would be a climbing fibre driven by retinal slip in a specific direction (*Graf et al., 1988*) or responding only to a small patch of skin (*Garwicz et al., 2002*). These receptive fields are quite stereotyped and have proven to be reliable landmarks in the functional regionalisation of the cerebellum; they are moreover tightly associated with the genetically specified zebrin patterning of the cerebellum (*Schonewille et al., 2006b*; *Mostofi et al., 2010*; *Apps and Hawkes, 2009*).

The apparently extreme specialisation of climbing fibres would limit the ability of the cerebellar circuitry to optimise complex movements. We can illustrate this with a human behaviour: riding a bicycle, which is often taken as an example of a typical cerebellar behaviour. This is an acquired skill for which there is little evolutionary precedent. It is likely to involve learning somewhat arbitrary arm movements in response to vestibular input (it is possible to ride a bike with one's eyes closed). The error signals guiding learning could be vestibular, visual or possibly cutaneous/nociceptive (as a result of a fall), but not necessarily those related to the arm whose movement is learnt. How can such disparate or uncommon but sometimes essential error signals contribute to cerebellar control of the arm? We call this the 'bicycle problem'.

At least two, non-exclusive solutions to this problem can be envisaged (see *Figure 11*). The first we term 'output convergence'; it involves the convergence of multiple cerebellar regions receiving climbing fibres of different modalities onto each specific motor element (for instance a muscle) being controlled. Striking, if partial, evidence for this is found in a study by (*Ruigrok et al., 2008*), who injected the retrograde, trans-synaptic tracer rabies virus into individual muscles. Multiple cerebellar

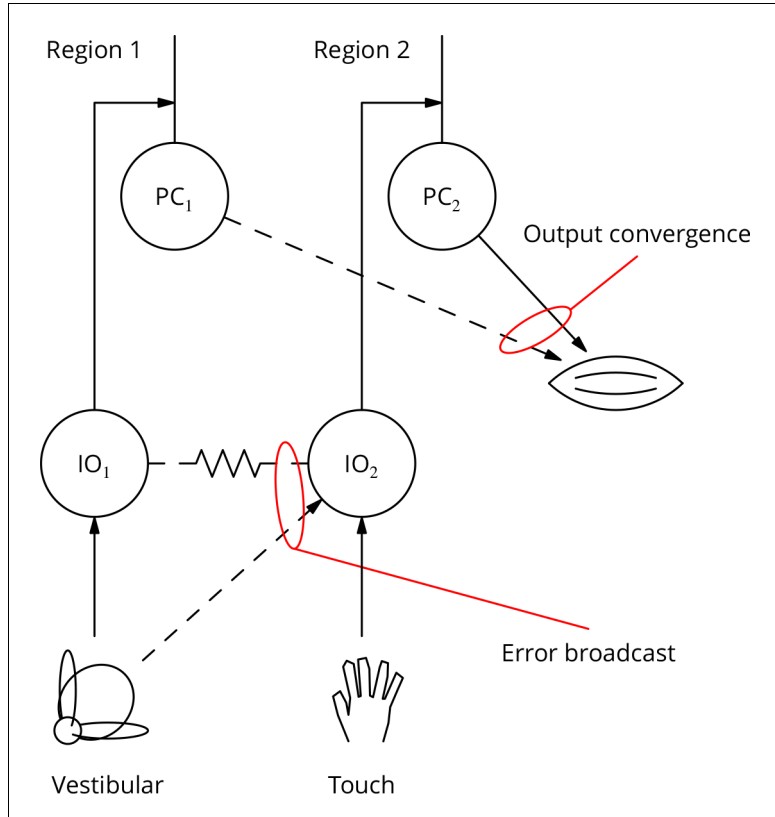

**Figure 11.** Diagram illustrating two possible solutions to the 'bicycle problem': how to use vestibular error information to guide learning of arm movements to ride a bicycle. In the 'output convergence' solution, the outputs from cerebellar regions receiving different climbing fibre modalities converge onto a motor unit (represented by a muscle in the diagram). In the 'error broadcast' solution, error complex spikes are transmitted beyond their traditional receptive fields, either by divergent synaptic inputs and/or via the strong electrical coupling between inferior olivary neurones.

DOI: https://doi.org/10.7554/eLife.31599.014

zones were labelled, showing that they all contribute to the control of those muscles, as posited. What is currently less clear is whether such separate zones receive climbing fibre inputs with different modalities. We note that the output convergence solution to the bicycle problem implies that the synaptic changes in those regions receiving the appropriate error information must outweigh any drift in synaptic weights from those regions deprived of meaningful error information. Adaptation of the learning rate, an algorithmic extension we suggest below, could contribute to ensuring that useful synaptic changes dominate.

We term the second solution to the bicycle problem the 'error broadcast' solution. According to this, error inputs to the olive are broadcast to olivary neurones (and Purkinje cells) outside the traditional receptive field. Although the weight of literature appears to be against this, there are both possible mechanisms and a small amount of supporting data for this suggestion. In terms of mechanism, the well-known electrical coupling of olivary neurones (*Devor and Yarom, 2002*) could recruit cells that do not directly receive suprathreshold synaptic input. This may occur much more frequently in vivo than in the quiescent/anesthetised conditions employed for most studies of climbing fibre receptive fields. Evidence for 'broadcast' of what we would term error complex spikes in vivo has been reported for visual and auditory stimuli (*Mortimer, 1975*; *Ozden et al., 2012*); these stimuli may be correlated with startle responses. Eye blink conditioning using a visual unconditioned stimulus has also been reported (*Rogers et al., 1999*).

The existence of broadcast error complex spikes would provide a mechanism explaining the giant IPSPs in the cerebellar nuclei elicited by peripheral stimulation (*Bengtsson et al., 2011*) and could

also account for the correlation between visual smooth pursuit behaviour and the nature of individual complex spikes (*Yang and Lisberger, 2014*): behaviour could only correlate with a single cell if others were receiving the same input.

## Possible extensions to the algorithm

Our implementation of cerebellar stochastic gradient descent and its simulation were purposefully kept as simple as possible, to provide a proof-of-concept with a minimum of assumptions and to simplify parallel analyses. It is likely that parts of the implementation will need to be altered and/or extended as further information becomes available.

Probably the most uncertain element of the implementation is the adaptive site in the cancellation of the average error. We chose to make the mossy fibre–nucleo-olivary neurone synapse plastic, but the plasticity could certainly operate in the olive instead of or in addition to the cerebellar nuclear site. Further studies of synaptic transmission and plasticity in both structures are clearly warranted in this context.

A simplification in our implementation is that it represents brief, discrete commands in what amounts to an offline learning rule. Error complex spikes are only emitted after the command and indeed were not simulated explicitly. This has the great advantage of avoiding the question of whether Purkinje cells that have not received a perturbation complex spike would interpret a broadcast error complex spike as a perturbation. A first remark is to stress the fact that the movement evaluation will often only occur after the movement is complete. Even if an eCS caused a small movement or some additional plasticity, that might be less of an issue outside of a movement context. Additionally, it is possible that cellular mechanisms could exist that would enable the Purkinje cell to distinguish the two types of input and therefore avoid spurious plasticity. The most obvious mechanism would be that, as already hinted at in the literature, error complex spikes are likely to be stronger. An extended synaptic plasticity rule could therefore include a case in which an error complex spike received in the absence of a recent perturbation spike has a neutral plasticity effect. There is currently little data on which to base a detailed implementation, although we note that in our experiments, bursts of 6 climbing fibre stimuli—a strong complex spike—resulted in only a modest LTP compared to the GP_ protocol.

A potentially unsatisfactory aspect of our simulations was the time taken to learn. Of the order of 50,000 iterations were required to optimise the 400 independent variables of the cerebellar output. Stochastic gradient descent is inherently slow, since just one or a few of those variables can be perturbed in one movement realisation, and the weight changes are furthermore individually small. Before considering possible acceleration methods, we note that some motor behaviours are repeated huge numbers of times. An obvious example is locomotion. Thus, public health campaigns in vogue in the USA at the time of writing aim for people to take 10,000 steps per day. So, clearly, a target of a few hundred thousand steps could be achieved in a matter of days or weeks.

Part of the slowness of learning results from the conflicting pressures on the plastic weight changes. Large changes allow rapid learning, but could prevent accurate optimisation. An obvious extension of our implementation that would resolve this conflict would be to allow large plastic changes far from the optimum but to reduce them as the optimum is approached. The information required to do this is available as the net drive (error excitation − cancellation inhibition) to the olivary neurones at the time of emission of an error complex spike. If the drive is strong, one can imagine a long burst of action potentials being emitted. There is in vitro (*Mathy et al., 2009*) and in vivo (*Yang and Lisberger, 2014*; *Rasmussen et al., 2013*) evidence that climbing fibre burst length can influence plasticity in Purkinje cells. It seems possible that the same could be true in the cerebellar nuclei (or alternative plastic site in the subtraction of the average error). However, the above mechanism for adapting learning rates would only work directly in the LTD direction, since olivary cells cannot signal the strength of a net inhibition when no error complex spike is emitted.

A mechanism that could regulate the speed of learning in both LTP and LTD directions would be to target perturbations to the time points where they would be most useful—those with the greatest errors. This might be achieved by increasing the probability (and possibly the strength) of the perturbation complex spikes shortly before strongly unbalanced (excitatory or inhibitory) inputs to the olive. This process offers a possible interpretation for various observations of complex spikes occurring before error evaluation: related to movements (*Bauswein et al., 1983*; *Kitazawa et al., 1998*) or triggered by conditioned stimuli (*Rasmussen et al., 2014*; *Ohmae and Medina, 2015*).

Movement-specific adaptations of the learning rates could provide an explanation for the phenomenon of 'savings', according to which relearning a task after extinction occurs at a faster rate than the initial learning. The adaptations could plausibly be maintained during extinction and therefore persist until the relearning phase. These adaptations could appear to represent memories of previous errors (*Herzfeld et al., 2014*).

Finally, the output convergence solution we proposed above for the bicycle problem could also reflect a parallelisation strategy enabling the computations involved in stochastic gradient descent to be scaled from the small circuit we have simulated to the whole cerebellum. As mentioned above, this might involve one of the above schemes for adjusting learning rates in a way that would help plasticity in regions with 'useful' error information to dominate changes in those without.

## Insight into learning in other brain regions

We believe that our proposed implementation of stochastic gradient descent offers possible insight into learning processes in other brain regions.

To date, the most compelling evidence for a stochastic gradient descent mechanism has been provided in the context of the acquisition of birdsong. A specific nucleus, the 'LMAN' has been shown to be responsible for song variability during learning and also to be required for learning (*Doya and Sejnowski, 1988*; *Olveczky et al., 2005*). Its established role is therefore analogous to our perturbation complex spike. Our suggestion that the same input signals both perturbation and error change may also apply in the birdsong context, where it would imply that LMAN also assumes the role of determining the sign of plasticity at the connections it perturbed. However, such an idea has for now not been examined and there is as yet a poor understanding of how the trial song is evaluated and of the mechanism for transmitting that information to the adaptive site; indeed the adaptive site itself has not been identified unequivocally.

We see a stronger potential analogy with our mechanism of stochastic gradient descent in the learning of reward-maximising action sequences by the basal ganglia. Under the stimulus of cortical inputs, ensembles of striatal medial spiny neurones become active, with the resulting activity partition determining the actions selected by disinhibition of central pattern generators through inhibition of the globus pallidus (*Grillner et al., 2005*). It is thought that the system learns to favour the actions that maximise the (discounted) reward, which is signalled by activity bursts in dopaminergic midbrain neurones and phasic release of dopamine, notably in the striatum itself (*Schultz, 1986*). This has been argued (*Schultz et al., 1997*) to reflect *reinforcement learning* or more specifically *temporal difference learning* (*Sutton and Barto, 1998*).

We note that temporal difference learning can be decomposed into two problems: linking actions to potential future rewards and a gradient ascent to maximise reward. In respect of the gradient ascent, we note that dopamine has a second, very well-known action in the striatum: it is necessary for the initiation of voluntary movements, since reduction of dopaminergic input to the striatum is the cause of Parkinson's disease, in which volitional movement is severely impaired. The key point is to combine the two roles of the dopaminergic system in the initiation of movement and in signalling reward. The initiation of movement by dopamine, which could contribute to probabilistic action selection, would be considered analogous to our perturbation complex spike and could create an eligibility trace. A subsequent reward signal would result in plasticity of eligible synapses reinforcing the selection of that action, with possible sites of plasticity including the cortico-striatal synapses that were successful in exciting an ensemble of striatal neurones (Rui Costa made a similar suggestion at the 5th Colloquium of the Institut du Fer à Moulin, Paris, 2014). This would constitute a mechanism of gradient ascent analogous to that we have proposed for gradient descent in the cerebellar system.

Of particular interest is whether correct optimisation also involves a mechanism for subtracting the average error in order to extract gradient information. Such subtraction would be entirely consistent with the reports of midbrain dopaminergic neurones responding more weakly to expected rewards and responding with sub-baseline firing to omission of a predicted reward. These phenomena moreover appear to involve an adaptive inhibitory mechanism (*Eshel et al., 2015*). These observations could be interpreted as a subtraction of the average reward by a process analogous to that we propose for the extraction of the change of error $\delta\mathcal{E}$.

## Conclusion

We have proposed a complete and plausible mechanism of stochastic gradient descent in the cerebellar system, in which the climbing fibre perturbs movements, creates an eligibility trace, signals error changes and guides plasticity at the sites of perturbation. We verify predicted plasticity rules that contradict the current consensus and highlight the importance of studying plasticity under physiological conditions. The gradient descent requires extraction of the change of error and we propose an adaptive inhibitory mechanism for doing this via cancellation of the average error. Our implementation of stochastic gradient descent suggests the operation of an analogous mechanism (of gradient ascent) in the basal ganglia initiated and rewarded by dopaminergic signalling.

# Materials and methods

## Electrophysiology

Animal experimentation methods were authorised by the 'Charles Darwin N°5' ethics committee (authorisation no. 4445). Adult female C57Bl/6 mice (2–5 months old) were anesthetised with isoflurane (Nicholas Piramal Ltd, India) and killed by decapitation. The cerebellum was rapidly dissected into cold solution containing (in mM): 230 sucrose, 26 NaHCO$_3$, 3 KCl, 0.8 CaCl$_2$, 8 MgCl$_2$, 1.25 NaH$_2$PO$_4$, 25 D-glucose supplemented with 50 $\mu$M D-APV to protect the tissue during slicing. 300 $\mu$m sagittal slices were cut in this solution using a Campden Instruments 7000 smz and stored at 32°C in a standard extracellular saline solution containing (in mM): 125 NaCl, 2.5 KCl, 1.5 CaCl$_2$, 1.8 MgCl$_2$, 1.25 NaH$_2$PO$_4$, 26 NaHCO$_3$ and 25 D-glucose, bubbled with 95 % O$_2$ and 5 % CO$_2$ (pH 7.4). Slices were visualised using an upright microscope with a 40 X, 0.8 NA water-immersion objective and infrared optics (illumination filter 750 $\pm$ 50 nm). The recording chamber was continuously perfused at a rate of 4–6 ml min$^{-1}$ with a solution containing (mM): 125 NaCl, 2.5 KCl, 1.5 CaCl$_2$, 1.8 MgCl$_2$, 1.25 NaH$_2$PO$_4$, 26 NaHCO$_3$, 25 D-glucose and 10 tricine, a Zn$^{2+}$ buffer (*Paoletti et al., 1997*), bubbled with 95 % O$_2$ and 5 % CO$_2$ (pH 7.4). Patch pipettes had resistances in the range 2–4 M$\Omega$ with the internal solutions given below. Unless otherwise stated, cells were voltage clamped at −70 mV in the whole-cell configuration. Voltages are reported without correction for the junction potential, which was about 10 mV (so true membrane potentials were more negative than we report). Series resistances were 4–10 M$\Omega$ and compensated with settings of ~90 % in a Multiclamp 700B amplifier (Molecular Devices). Whole-cell recordings were filtered at 2 kHz and digitised at 10 kHz. Experiments were performed at 32–34°C. The internal solution contained (in mM): 128 K-gluconate, 10 HEPES, 4 KCl, 2.5 K$_2$HPO$_4$, 3.5 Mg-ATP, 0.4 Na$_2$-GTP, 0.5 L-(–)-malic acid, 0.008 oxaloacetic acid, 0.18 $\alpha$-ketoglutaric acid, 0.2 pyridoxal 5'-phosphate, 5 L-alanine, 0.15 pyruvic acid, 15 L-glutamine, 4 L-asparagine, 1 reduced L-glutathione, 0.5 NAD$^+$, 5 phosphocreatine, 1.9 CaCl$_2$, 1.5 MgCl$_2$, 0.1 K$_{3.8}$EGTA. Free [Ca$^{2+}$] was calculated with Maxchelator (C. Patton, Stanford) to be 120 nM. Chemicals were purchased from Sigma-Aldrich, D-APV from Tocris.

Recordings were made in the vermis of lobules three to eight of the cerebellar cortex. Granule cell EPSCs were elicited with stimulation in the granule cell layer with a glass pipette of tip diameter 8–12 $\mu$m filled with HEPES-buffered saline. Climbing fibre electrodes had $\sim 2\,\mu$m diameter and were also positioned in the granule cell layer. Images were taken every 5 min; experiments showing significant slice movement (>20 $\mu$m) were discarded. Stimulation intensity was fixed at the beginning of the experiment (1–15 V; 50–200 $\mu$s) and maintained unchanged during the experiment.

## Analysis

No formal power calculation to determine the sample sizes was performed for the plasticity experiments. Recordings were analysed from 75 cells in 55 animals. Slices were systematically changed after each recording. Animals supplied 1–3 cells to the analysis, usually with rotating induction protocols, but there was no formal randomisation or blocking and there was no blinding. An initial analysis focusing on between-protocol comparisons was performed when $n = 33$ cells were retained for analysis (criteria detailed below). Because the p-value for the GPE vs. G_E difference was close to a $p<0.001$ threshold (*Colquhoun, 2014*) and the confidence interval for the GPE plasticity ratio was narrower than for the others, acquisition for that protocol was reduced and additional experiments prioritised the G_ _, GP_ and G_E protocols. Experiments were ultimately halted for external reasons, when $n = 58$ cells were retained. The two-stage analysis is a form of multiple comparison; in

consequence, the output of the statistical test on the final data has been corrected by doubling the indicated p-values in *Table 1*.

Inspection of acquired climbing fibre responses revealed some failures of stimuli after the first in a fraction of cells, presumably because the second and subsequent stimuli at short intervals fell within the relative refractory period. As a complex spike was always produced these cells have been included in our analysis, but where individual data are displayed, we identify those cells in which failures of secondary climbing fibre stimuli were observed before the end of the induction period.

Analysis made use of a modular Python framework developed in house. Analysis of EPSC amplitudes for each cell began by averaging all of the EPSCs acquired to give a smooth time course. The time of the peak of this 'global' average response was determined. Subsequent measurement of amplitudes of other averages or of individual responses was performed by averaging the current over 0.5 ms centred on the time of the peak of the global average. The baseline calculated over 5 ms shortly before the stimulus artefact was subtracted to obtain the EPSC amplitude. Similar analyses were performed for both EPSCs of the paired-pulse stimulation.

Individual EPSCs were excluded from further analysis if the baseline current in the sweep exceeded −1 nA at −70 mV. Similarly, the analysis automatically excluded EPSCs in sweeps in which the granule cell stimuli elicited an action potential in the Purkinje cell (possibly through antidromic stimulation of its axon or through capacitive coupling to the electrode). However, during induction, in current clamp, such spikes were accepted. For displaying time series, granule cell responses were averaged in bins of 2 min.

The effects of series resistance changes in the Purkinje cell were estimated by monitoring the transient current flowing in response to a voltage step. The amplitude of the current 2 ms after the beginning of the capacity transient was measured. We shall call this the 'dendritic access'. Modelling of voltage-clamp EPSC recordings in a two-compartment model of a Purkinje cell (*Llano et al., 1991*) suggests that this measure is approximately proportional to EPSC amplitude as the series resistance changes over a reasonable range (not shown). It therefore offers a better estimate of the effect on the EPSC amplitude of series resistance changes than would the value of the series resistance (or conductance), which is far from proportional to EPSC amplitude. Intuitively, this can be seen to arise because the EPSC is filtered by the dendritic compartment and the measure relates to the dendritic component of the capacitive transient, whereas the series resistance relates to the somatic compartment. We therefore calculated $R_{res}$, the ratio of the dendritic access after induction (when plasticity was assessed) to the value before induction, in order to predict the changes the EPSC amplitude arising from changes of series resistance.

Because we elicited EPSCs using constant-voltage stimulation, variations of the resistance of the tip of the stimulating electrode (for instance if cells are drawn into it) could alter the stimulating current flow in the tissue. We monitored this by measuring the amplitude of the stimulus artefact. Specifically, we calculated $R_{stim}$, the after/before ratio of the stimulation artefact amplitude.

We then used a robust linear model to examine the extent to which changes of series resistance or apparent stimulation strength could confound our measurements of plasticity, which we represented as the after/before ratio of EPSC amplitudes $R_{EPSC}$; the model (in R syntax) was:

$$R_{EPSC} \sim \mathrm{protocol} + R_{res} + R_{stim}$$

This showed that series resistance changes, represented by $R_{res}$, had a significant influence (*t*-value 2.30, 69 degrees of freedom) with a slope close to the predicted unity (1.14). In contrast, changes of the stimulus artefact had no predictive value (slope −0.003, *t*-value −0.01).

Extending our modelling using a mixed-effect model (call 'lmer' in the lme4 R package) to include animals as a random effect did not indicate that it was necessary to take into account any between-animal variance, as this was reported to be zero. On this basis, we consider each cell recorded to be the biological replicate.

We did not wish to rely on the parametric significance tests of the linear model for comparing the plasticity protocols (although all of comparisons we report below as significant were also significant in the model). Instead, we equalised the dendritic filtering and stimulation changes between groups by eliminating those cells in which $R_{res}$ or $R_{stim}$ differed by more than 20% from the mean values for all cells (0.94 ± 0.10 and 1.01 ± 0.19, respectively; mean ± sd, $n = 75$). After this operation, which eliminated 17 cells out of 75 leaving 58 (from 47 animals), the mean ratios varied by only a few

percent between groups (ranges 5 % and 2 % for $R_{res}$ and $R_{stim}$, respectively) and would be expected to have only a minimal residual influence. Normalising the $R_{EPSC}$s of the trimmed groups by $R_{res}$ did not alter the conclusions presented below. Note that after this trimming, the remaining changes of $R_{res}$ imply that all EPSC amplitudes after induction were underestimated by about 6 % relative to those at the beginning of the recording. The differences of $R_{EPSC}$ between induction protocols were evaluated statistically using two-tailed nonparametric tests implemented by the 'wilcox. test' command in R (*R Core Team, 2013*).

95 % confidence limits were calculated using R bootstrap functions ('BCa' method). For confidence limits of differences between means, stratified resampling was used.

A small additional series of experiments of 16 cells from 16 animals of which 12 were retained by similar criteria to those explained above (except that series resistance changes had to be evaluated using the current amplitude 1 ms into the response to the voltage step) were analysed separately and underlie the plasticity data in *Figure 6—figure supplement 1*.

## Simulation methods

Network simulations were designed as follows (see also diagram in *Figure 12*). A total of $S \times L$ Purkinje cells were placed on a rectangular grid of extent $S$ in the sagittal plane and width $L$ in the lateral direction. The activity of each Purkinje cell during a 'movement' was characterised by its firing rate in $T$ time bins $\{t = 1, \cdots, T\}$.

$$\{PC_{s,l}(t), s = 1, \cdots, S, l = 1, \cdots, L\} \tag{1}$$

Purkinje cells contacted projection neurones ($PN$) and nucleo-olivary neurones ($NO$) in the cerebellar nuclei that contained $L$ cells of each type. The activities of both types of nuclear cell were also characterised by their firing rates in the different time bins, $\{PN_l(t), l = 1, \cdots, L\}$ and $\{NO_l(t), l = 1, \cdots, L\}$. Mossy fibres, granule cells (parallel fibres) and molecular layer interneurons were subsumed into a single cell type $M$ (for mossy fibre) with $N$ cells restricted to each row of $L$ Purkinje cells, with a total of $N \times S$ mossy fibres. Mossy fibre activity was represented in a binary manner, $M_{i,s}(t) = 0$ or 1, with $\{i = 1, \cdots, N\}$ and $\{s = 1, \cdots, S\}$.

The connectivity was chosen such that all Purkinje cells in the sagittal 'column' $l$ projected to the $l$-th nuclear projection neurone with identical inhibitory (negative) weights, as well as to the $l$-th nucleo-olivary neurone with potentially different, but also identical inhibitory weights. Mossy fibres were chosen to project to Purkinje cells with groups of $N$ fibres at a given sagittal position $s$, contacting cells on the row of $L$ Purkinje cells at the sagittal position with probability 1/2. The activity of Purkinje and cerebellar nuclear cells in the absence of climbing fibre activity was thus

$$PC_{s,l}(t) = \Phi\left(\sum_{i=1,N} w^l_{i,s} \sigma^l_{i,s} M_{i,s}(t)\right) \tag{2}$$

$$PN_l(t) = \Phi\left(\sum_{i=1,N;s=1,S} u^l_{i,s} M_{i,s}(t) + u_{(PC \to PN)} \sum_{s=1,S} PC_{s,l}(t)\right) \tag{3}$$

$$NO_l(t) = \Phi\left(\sum_{i=1,N;s=1,S} v^l_{i,s} M_{i,s}(t) + u_{(PC \to NO)} \sum_{s=1,S} PC_{s,l}(t)\right), \tag{4}$$

where $\sigma^l_{i,s}$ enforces the 1/2 probability of connection between a Purkinje cell and a parallel fibre that traverses its dendritic tree: $\sigma^l_{i,s}$ is equal to 1 or 0 with probability 1/2, independently drawn for each triple $(i,s,l)$. The f-I curve $\Phi$ was taken to be a saturating threshold linear function, $\{\Phi(x) = 0 \text{ for } x < 0, \Phi(x) = x \text{ for } 0 < x < r_{max} \text{ and } \Phi(x) = r_{max} \text{ for } x > r_{max}\}$. The weights $u^l_{i,s}$ were non-plastic and chosen such that a given mossy fibre $(i,s)$ contacted a single projection neurone and a single nucleo-olivary neurone among the $L$ possible ones of each type. In other words, for each mossy fibre index $(i,s)$, a number $l_{(i,s)}$ was chosen at random with uniform probability among the $L$ numbers $\{1, \cdots, L\}$ and the weights $\{u^l_{i,s}, l = 1, \cdots, l\}$ were determined as

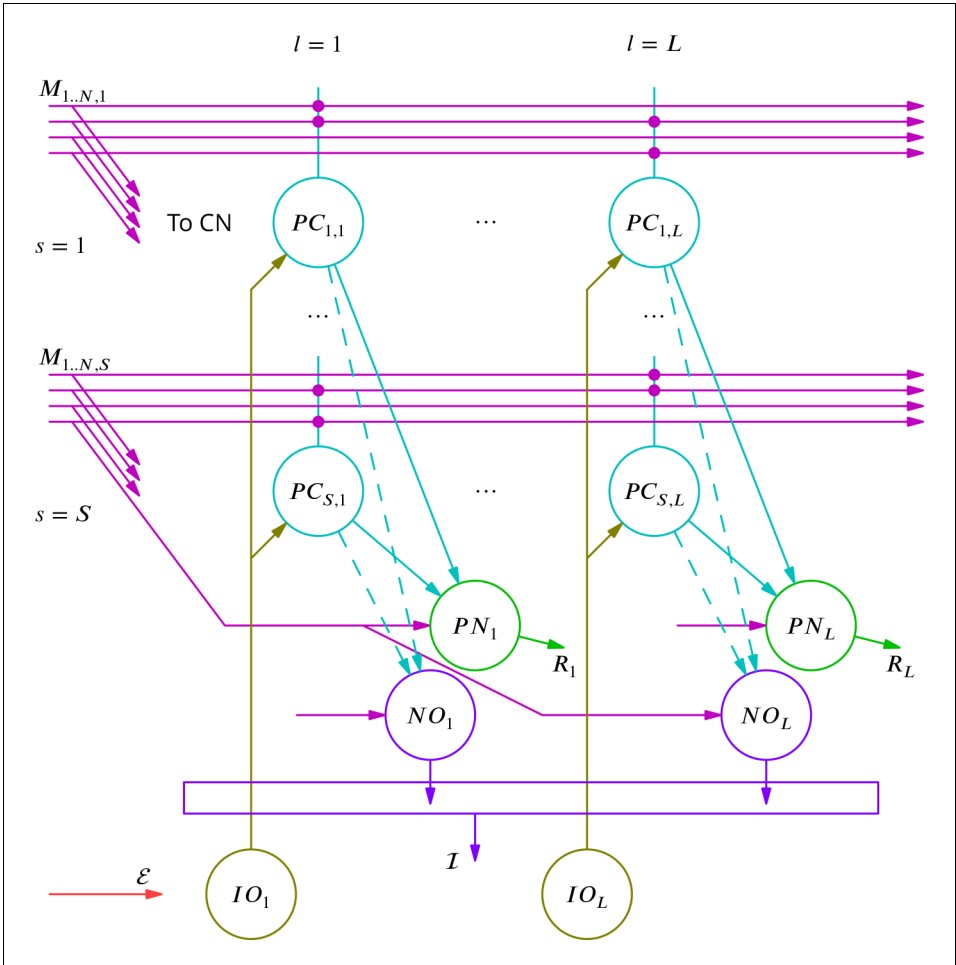

**Figure 12.** Diagram of the simulated network model. The details are explained in the text.
DOI: https://doi.org/10.7554/eLife.31599.015

$$u_{i,s}^{l} = \delta_{l,l_{(i,s)}} u_{(M \to PN)}, \tag{5}$$

where $u_{(M \to PN)}$ was a constant. The weights $v_{i,s}^{l}$ and $w_{i,s}^{l}$ were plastic and followed the learning dynamics described below (see *Equations 11 and 13*).

The learning task itself consisted of producing, in response to $\mu = 1, \cdots, p$ spatiotemporal patterns of mossy fibre inputs $M_{i,s}^{\mu}(t)$, the corresponding output target rates of the projection neurones $\left\{ R_l^{\mu}(t) \, l = 1, \cdots, L, \, t = 1, ..., T \right\}$. For each pattern $\mu$, the inputs were obtained by choosing at random with uniform probability $NS/2$ active fibres. For each active fibre $(i, s)$, a time bin $t_{(i,s)}^{\mu}$ was chosen at random with uniform probability and the activity of the fibre was set to one in this time bin, $M_{(i,s)}^{\mu}(t) = \delta_{t,t_{(i,s)}^{\mu}}$. The activity was set to zero in all time bins for the $NS/2$ inactive fibres. The target rates where independently chosen with uniform probability between 0 and $2\bar{r}_D$ for each projection neurone in each pattern $\mu$, where $\bar{r}_D$ is the desired average firing rate for both projection and nucleo-olivary neurones in the cerebellar nuclei.

The olivary neurones were not explicitly represented. It was assumed that the $L \times S$ Purkinje cells were contacted by $L$ climbing fibres with one climbing fibre contacting the $S$ Purkinje cells at a given lateral position.

The learning algorithm then proceeded as follows. Patterns $\left\{ M_{i,s}^{\mu}(t) \right\}$ were presented sequentially. After pattern $\mu$ was chosen, perturbations of Purkinje cell firing by complex spikes were

generated as follows. The probability that each climbing fibre emitted a perturbation complex spike was taken to be $\rho$ per pattern presentation; when a climbing fibre was active, it was considered to perturb the firing of its Purkinje cells in a single time bin chosen at random. Denoting by $\eta_l(t) = 1$ that climbing fibre $l$ had emitted a spike in time bin $t$ (and $\eta_l(t) = 0$ when there was no spike), the $S$ firing rates of the Purkinje cell at position $l$ (see *Equation 3*) were taken to be

$$PC_{s,l}(t) = \Phi\left(\sum_{i=1,N} w_{i,s}^l \sigma_{i,s}^l M_{i,s}(t) + \eta_l(t)A\right), s = 1, \cdots, S, \tag{6}$$

where $A$ defines the amplitude of the complex-spike perturbation of Purkinje cell firing.

Given the pattern $\mu$ and the firing of Purkinje cells (*Equation 6*), the activities of cerebellar nuclear neurones were given by *Equations 3 and 4*. The current 'error' for pattern/movement $\mu$ was quantified by the average distance of the projection neurones' activity from their target rates

$$\mathcal{E}^\mu = \frac{1}{LT}\sum_{l=1,L;t=1,T} |PN_l^\mu(t) - R_l^\mu(t)|. \tag{7}$$

The learning step after the presentation of pattern $\mu$ was determined by the comparison (not explicitly implemented) in the olivary neurones between the excitation $\mathcal{E}^\mu$ and the inhibition $\mathcal{I}^\mu$ coming from the discharges of nucleo-olivary neurones with

$$\mathcal{I}^\mu = \frac{1}{LT}\sum_{l=1,L;t=1,T} NO_l^\mu(t). \tag{8}$$

An error complex spike was propagated to all Purkinje cells after a 'movement' when the olivary activity $IO = \mathcal{E}^\mu - \mathcal{I}^\mu$ was positive. Accordingly, modifications of the weights of mossy fibre synapses on perturbed Purkinje cells ($w$) and on nucleo-olivary neurones ($v$) were determined after presentation of pattern $\mu$ by the sign $c$ of $IO$, $c = \text{sign}(IO)$, as,

$$w_{i,s}^l \rightarrow w_{i,s}^l - \alpha_w c \sum_{t=1,T} \eta_l(t) M_{i,s}^\mu(t) \tag{9}$$

$$v_{i,s}^l \rightarrow \left[v_{i,s}^l + \alpha_v c \sum_{t=1,T} M_{i,s}^\mu(t)\right]_+ \tag{10}$$

where the brackets served to enforce a positivity constraint on the weights $v_{i,s}^l$ ($[x]_+ = x, x > 0$ and $[x]_+ = 0, x < 0$).

In the reported simulations, the non-plastic weights in *Equations 3–5* were identical and constant for all synapses of a given type:

$$u_{(M \rightarrow PN)} = 4TL\bar{r}_D/NS \tag{11}$$

$$u_{(PC \rightarrow PN)} = -\bar{r}_D/(\bar{r}_{PC}S) \tag{12}$$

$$u_{(PC \rightarrow NO)} = qu_{(PC \rightarrow PN)} = -q\bar{r}_D/(\bar{r}_{PC}S), \tag{13}$$

where $q$ describes the strength of the $PC \rightarrow NO$ connection relative to the $PC \rightarrow PN$ connection. The initial weights of the plastic synapses were drawn from uniform distributions such that initial firing rates were on average given by $\overline{PC}_0 = \bar{r}_{PC}$ and $\overline{NO}_0 = 15$ Hz for Purkinje cells and nucleo-olivary neurones, respectively. The latter value was chosen such that the expected initial mismatch between inhibition $\mathcal{I}$ and excitation $\mathcal{E}$ for random pairings of initial $PN$ firing rates and target rates $R$ was of the order of 1 Hz. Consequently, for $M \rightarrow PC$ synapses $w_{i,s}^l$ was drawn from $[0, 8T\bar{r}_{PC}/N]$ and for $M \rightarrow NO$ synapses, weights $v_{i,s}^l$ were drawn from $\left[0, (\overline{NO}_0 + q\bar{r}_D)4TL/(NS)\right]$. These weights furthermore ensured initial average firing rates close to $\bar{r}_D$ in the nuclear projection neurones. Target rates $R$ are uniformly distributed between 0 Hz and $2\bar{r}_D$.

The parameters used in the reported simulation (full and simplified) are provided in *Table 2*.

In the following we describe the simplified version of the model without perturbations used to highlight shortcomings of the current Marr-Albus-Ito theory. $M$, $PC$, and $PN$ cell types are defined as above with identical synaptic connections and identical initial weights. The learning rules are different as no comparison between an estimated error (inhibition) and the global error (excitation) is made. We therefore do not consider the population of $NO$ cells in this version. Two methods of calculating the error were tested. In the first, the error conveyed by the climbing fibres contains information about the sign of the movement error and is given by

$$\mathcal{E}^{\mu} = \frac{1}{LT} \sum_{l=1,L;t=1,T} \left( R_l^{\mu}(t) - PN_l^{\mu}(t) \right). \tag{14}$$

In the second, the absolute values according to *Equation 7* were used.

The synaptic weight changes are as follows. Whenever the error is positive, the synaptic weights of active $M \rightarrow PC$ connections are decreased by an amount $\alpha_w$, while otherwise there is a slow positive drift of $M \rightarrow PC$ synaptic weights with increments $\beta_w$:

$$w_{i,s}^l \rightarrow \begin{cases} w_{i,s}^l - \alpha_w \sum_{t=1,T} M_{i,s}^{\mu}(t) & \text{if } \mathcal{E}^{\mu} > 0 \\ w_{i,s}^l + \beta_w & \text{else.} \end{cases} \tag{15}$$

In order to show relaxation of the average of $PN$ firing rates to the average of the target rates $R$, we chose different initial distributions of uniformly distributed target rates with averages between 10 Hz and 50 Hz.

We show in the main text that the four key parameters governing convergence of our learning algorithm are the learning rates of mossy fibre–Purkinje cell synapses $\alpha_w$ and mossy fibre–nucleo-olivary neurone synapses $\alpha_v$, as well as the probability of a perturbation complex spike occurring in a given movement in a given cell $\rho$, and the resulting amplitude of the perturbation of Purkinje cell firing $A$. Varying each of these $\pm$ 10 % individually altered the final error (1.4 Hz, averaged over trials 55,000–60,000) by at most 7%, indicating that this final output was not ill-conditioned or finely tuned with respect to these parameters.

The simulation was coded in Python.

**Table 2.** Parameters used in the simulation shown in *Figure 12*.

| Parameter | Symbol | Value |
|---|---|---|
| Sagittal extent | $S$ | 10 |
| Lateral extent | $L$ | 40 |
| Time bins per movement | $T$ | 10 |
| Mossy fibres per sagittal position | $N$ | 2000 |
| Maximum firing rate (all neurones) | $r_{max}$ | 300 Hz |
| pCS probability per cell per movement | $\rho$ | 0.03 |
| Amplitude of Purkinje cell firing perturbation | $A$ | 2 Hz |
| Learning rate of $M \rightarrow PC$ synapses | $\alpha_w$ | 0.02 |
| Learning rate of $M \rightarrow NO$ synapses | $\alpha_v$ | 0.0002 |
| Mean Purkinje cell firing rate | $\bar{r}_{PC}$ | 50 Hz |
| Mean firing rate for nuclear neurones | $\bar{r}_D$ | 30 Hz |
| $M \rightarrow PN$ synaptic weight | $u_{(M \rightarrow PN)}$ | 2.4 |
| $PC \rightarrow PN$ synaptic weight | $u_{(PC \rightarrow PN)}$ | −0.06 |
| $PC \rightarrow NO$ relative synaptic weight | $q$ | 0.5 |
| Weight increment of $M \rightarrow PC$ drift (simple model) | $\beta_w$ | 0.002 |

DOI: https://doi.org/10.7554/eLife.31599.016

## Acknowledgements

We are grateful to the following for discussion and/or comments on this manuscript: David Attwell, Mariano Casado, Paul Dean, Anne Feltz, Richard Hawkes, Clément Léna, Steven Lisberger, Tom Ruigrok, John Simpson, Brandon Stell, Stéphane Supplisson and German Szapiro. We thank Gary Bhumbra for sharing a native-python library for reading Clampex files.
 A preprint describing this work was posted on the bioRxiv repository on 2016-05-16.

## Additional information

### Funding

| Funder | Grant reference number | Author |
| --- | --- | --- |
| Agence Nationale de la Recherche | ANR-08-SYSC-005 | Boris Barbour |
| National Science Foundation | IIS-1430296 | Johnatan Aljadeff Nicolas Brunel |
| Fondation pour la Recherche Médicale | DEQ20160334927 | Boris Barbour |
| Fondation pour la Recherche Médicale | | Guy Bouvier |
| Région Ile-de-France | | Guy Bouvier |
| Labex | ANR-10-LABX-54 MEMOLIFE | Guy Bouvier Boris Barbour |
| Deutsche Forschungsgemeinschaft | RA-2571/1-1 | Jonas Ranft |
| Idex PSL* Research University | ANR-11-IDEX-0001-02 | Vincent Hakim Boris Barbour |

The funders had no role in study design, data collection and interpretation, or the decision to submit the work for publication.

### Author contributions

Guy Bouvier, Data curation, Investigation, Methodology, Writing—original draft, Writing—review and editing, Performed all experiments shown; Johnatan Aljadeff, Jonas Ranft, Software, Formal analysis, Writing—original draft, Writing—review and editing; Claudia Clopath, Writing—review and editing, Contributed to Initial theoretical exploration; Célian Bimbard, Investigation, Performed pilot experiments and suggested implementation of tracking plasticity; Antonin Blot, Software; Jean-Pierre Nadal, Formal analysis; Nicolas Brunel, Vincent Hakim, Conceptualization, Software, Formal analysis, Supervision, Writing—original draft, Writing—review and editing; Boris Barbour, Conceptualization, Data curation, Software, Formal analysis, Supervision, Funding acquisition, Methodology, Writing—original draft, Project administration, Writing—review and editing, Performed all data analysis shown

### Author ORCIDs

Guy Bouvier https://orcid.org/0000-0002-6160-7186
Johnatan Aljadeff http://orcid.org/0000-0002-7145-0514
Claudia Clopath http://orcid.org/0000-0003-4507-8648
Célian Bimbard http://orcid.org/0000-0002-6380-5856
Jonas Ranft http://orcid.org/0000-0002-7843-7443
Antonin Blot https://orcid.org/0000-0002-1546-3927
Jean-Pierre Nadal http://orcid.org/0000-0003-0022-0647
Nicolas Brunel https://orcid.org/0000-0002-2272-3248
Vincent Hakim http://orcid.org/0000-0001-7505-8192
Boris Barbour http://orcid.org/0000-0002-1911-0539

## Ethics

Animal experimentation: Animal experimentation methods were performed according to authorisation 04445.02 granted by the 'Charles Darwin N°5' ethics committee.

## Decision letter and Author response

Decision letter https://doi.org/10.7554/eLife.31599.025
Author response https://doi.org/10.7554/eLife.31599.026

---

## Additional files

### Supplementary files

• Transparent reporting form
DOI: https://doi.org/10.7554/eLife.31599.017

### Data availability

Source data, analysis/simulation scripts and software libraries have been depositied at the Zenodo repository.

The following datasets were generated:

| Author(s) | Year | Dataset title | Dataset URL | Database and Identifier |
|---|---|---|---|---|
| Guy Bouvier, Johnatan Aljadeff, Claudia Clopath, Célian Bimbard, Jonas Ranft, Antonin Blot, Jean-Pierre Nadal, Nicolas Brunel, Vincent Hakim, Boris Barbour | 2018 | Cerebellar learning using perturbations: data, analysis/simulation scripts | https://doi.org/10.5281/zenodo.1481929 | Zenodo, 10.5281/zenodo.1481929 |
| Guy Bouvier, Johnatan Aljadeff, Claudia Clopath, Célian Bimbard, Jonas Ranft, Antonin Blot, Jean-Pierre Nadal, Nicolas Brunel, Vincent Hakim, Boris Barbour | 2018 | Cerebellar learning using perturbations: software libraries | https://doi.org/10.5281/zenodo.1481925 | Zenodo, 10.5281/zenodo.1481925 |

---

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

## Appendix 1

DOI: https://doi.org/10.7554/eLife.31599.018

# Mathematical appendix

## Reduced model

The model focuses on the rate $P(\tau)$ of one P-cell. Learning a pattern (movement) requires adjustment of the $P(\tau)$ rate to the target value $R$. The movement error is defined as

$$\mathcal{E}(\tau) = |P(\tau) - R| \qquad (16)$$

Note that here $\tau$ represents trial/learning step number rather than temporal variations within a single movement realisation. In this simplified model, the P-cell activity during a movement is characterised by a single value rather than a sequence of values in time.

The other variable of the model besides $P(\tau)$ is $\mathcal{J}(\tau)$. It is used to quantify the strength

$$\mathcal{I}(\tau) = [\mathcal{J}(\tau) - qP(\tau)]_+ \qquad (17)$$

of the $(NO \rightarrow \mathcal{I}O)$ inhibitory input which provides the current value of the estimated global error. The two terms in the expression of the current are meant to represent the excitatory action of the MF inputs on the discharge of the $NO$ ($\mathcal{J}(\tau)$) and the inhibitory effects of the Purkinje cell discharge ($-qP(\tau)$). Learning steps consist of updates of the values $P(\tau)$ and $\mathcal{J}(\tau)$. They are of two kinds depending on whether the P-cell rate is perturbed or not.

Updates of type $\Delta P \Delta \mathcal{J}$ proceed as follows. The firing rate of the P-cell is increased by $A > 0$ and becomes $P(\tau) + A$. The value of the global error corresponding to this perturbed firing rate is thus

$$\mathcal{E}_p = |P(\tau) + A - R| \qquad (18)$$

$P(\tau)$ and $\mathcal{J}(\tau)$ are updated depending on whether $\mathcal{E}_p$ is smaller or greater than the current value of the estimated global error $[\mathcal{J}(\tau) - qP(\tau) - qA]_+$. The parameter $q > 0$ quantifies how much the $NO$ output is affected by the P-cell discharge, with the brackets indicating rectification (which imposes the constraint that the $NO$ output remains non-negative).

- If $\mathcal{E}_p > [\mathcal{J}(\tau) - qP(\tau) - qA]_+$ the perturbation is judged to have increased the error and therefore to have the wrong sign: the perturbed firing rate needs to be decreased. Concurrently, $\mathcal{J}$ needs to be increased since $\mathcal{I}$ is judged too low compared to the real value of the error. Thus $P(\tau)$ and $\mathcal{J}(\tau)$ are changed to

$$P(\tau + 1) = [P(\tau) - \Delta P]_+ \qquad (19)$$

$$\mathcal{J}(\tau + 1) = \mathcal{J}(\tau) + \Delta \mathcal{J} \qquad (20)$$

As above the brackets indicate rectification, which imposes the constraint that firing rates are non-negative.

- If $\mathcal{E}_p < [\mathcal{J}(\tau) - qP(\tau) - qA]_+$, the converse reasoning leads to changes of $P(\tau)$ and $\mathcal{J}(\tau)$ in the opposite directions

$$P(\tau + 1) = P(\tau) + \Delta P \qquad (21)$$

$$\mathcal{J}(\tau + 1) = [\mathcal{J}(\tau) - \Delta \mathcal{J}]_+ \qquad (22)$$

Updates of type $\Delta\mathcal{J}$ are performed as described above but without any perturbation ($A = 0$) and without any update of $P$ (only the estimated error $\mathcal{I}(\tau)$ is updated in updates of type $\Delta\mathcal{J}$). Namely,

- If $\mathcal{E} > [\mathcal{J}(\tau) - qP(\tau)]_+$ the estimated error is judged too low compared to the real value of the error and $\mathcal{J}$ needs to be increased. Thus $P(\tau)$ is kept unchanged and $\mathcal{J}(\tau)$ is modified

$$P(\tau+1) = P(\tau) \tag{23}$$

$$\mathcal{J}(\tau+1) = \mathcal{J}(\tau) + \Delta\mathcal{J} \tag{24}$$

- If $\mathcal{E} < [\mathcal{J}(\tau) - qP(\tau)]_+$, the converse reasoning leads to changes $\mathcal{J}(\tau)$ in the opposite directions

$$P(\tau+1) = P(\tau) \tag{25}$$

$$\mathcal{J}(\tau+1) = [\mathcal{J}(\tau) - \Delta\mathcal{J}]_+ \tag{26}$$

The proposed core operation of the algorithm for one movement can thus be described as follows. At each time step, a perturbation occurs with probability $\rho$. The occurrence of a perturbation gives rise to an update of type $\Delta P \Delta \mathcal{J}$, performed as described by **Equations 19-22**. For the complementary $(1 - \rho)$ fraction of time steps, no perturbation occurs ($A = 0$) and updates of type $\Delta\mathcal{J}$ are performed, as described by **Equations 23-26**.

This abstract model depends on five parameters: $A$, the amplitude of the rate perturbation, $\Delta P$ and $\Delta \mathcal{J}$, the update amplitudes of the rate and error estimate, respectively, $q$ the strength of inhibition of the NO by the P-cell, and $\rho$, which describes the probability of an update of type $\Delta P \Delta \mathcal{J}$ (and the complementary probability $1 - \rho$ of $\Delta \mathcal{J}$ updates). The ratio of the update amplitudes play an important role and is denoted by $\beta$, $\beta = \Delta\mathcal{J}/\Delta P$. The target firing rates are chosen in the range $[0, R_{max}]$, which simply fixes the firing rate scale. We would like to determine the conditions on the five parameters $A, \Delta P, \Delta\mathcal{J}, q$ and $\rho$ for the algorithm to 'converge'. For simplicity, we here consider only constant perturbation and learning steps. Therefore, at best the rate and the error fluctuates around the target rates and zero error, in a bounded domain of size determined by the magnitude of the constant perturbation and learning steps. We say that the algorithm converges when this situation is reached. We would also like to understand how they determine the rate of convergence and the residual error.

## Convergence in the one-cell case

Stochastic gradient descent is conveniently analysed with a phase-plane description, by following the values $(P, \mathcal{J})$ from one update to the next in the $(P, \mathcal{J})$ plane. The dynamics randomly alternate between updates of type $\Delta P \Delta \mathcal{J}$ and $\Delta \mathcal{J}$, depending on whether the P-cell rate is perturbed by the CF or not. We consider these two types of update in turn.

For updates of type $\Delta P \Delta \mathcal{J}$, the update depends on whether the perturbed error $\mathcal{E}_p = |P - R + A|$ is larger or smaller than the estimated error $[\mathcal{J} - qP]_+$. Namely, it depends on the location of the current $(P, \mathcal{J})$ with respect to the two lines in the $P$-$\mathcal{J}$ plane, $C_\pm$, $\{(q \pm 1)P + (q \pm 1)A \mp R = \mathcal{J}\}$, of slopes $q \pm 1$ (see **Figure 9B**). The dynamics of **Equations 19-22** are such that each update moves the point $(P, \mathcal{J})$ by adding to it the vectorial increment $\pm(\Delta P, -\Delta\mathcal{J})$ with the $+$ sign holding in the quadrant above the lines $C_\pm$ and the minus sign elsewhere. These updates move the point $(P, \mathcal{J})$ towards the lines $C_\pm$. In the triangular domain below the line $C_-$, the update does not directly move the $(P, \mathcal{J})$ trajectory towards the $C_-$ line, when $\beta < (1 - q)$ and the update has an angle greater than the inclination of the $C_-$. However, it reduces $P$. When $P = 0$ below $C_-$, the updates are strictly upward and towards $C_-$.

Updates of type $\Delta\mathcal{J}$ depend on the location of the point $(P, \mathcal{J})$ with respect to the lines $D_\pm$, $\{(q \pm 1)P \mp R = \mathcal{J}\}$, since the cell firing rate is not perturbed in these updates. In the quadrant above the lines $D_\pm$, an update moves the point $(P, \mathcal{J})$ downwards by $\Delta\mathcal{J}$,

in other words, adds the vectorial increment $(0, -\Delta\mathcal{J})$. In the complementary domain of the $(P, \mathcal{J})$ plane, an update moves the point $(P, \mathcal{J})$ upwards by $\Delta\mathcal{J}$, in other words, adds the opposite vectorial increment $(0, \Delta\mathcal{J})$. Both updates move $(P, \mathcal{J})$ towards the lines $D_\pm$.

Learning proceeds by performing updates of type $\Delta P\Delta\mathcal{J}$ and $\Delta\mathcal{J}$ with respective probabilities $\rho$ and $(1-\rho)$. Starting from an initial coordinate $(P, \mathcal{J})$, three phases can be distinguished as described in the main text.

Above the lines $C_+$ and $D_-$, the mixed updates lead the point $(P, \mathcal{J})$ to perform a random walk with a systematic mean rightward-downward drift per update equal to

$$\rho(\Delta P, -\Delta\mathcal{J}) + (1-\rho)(0, -\Delta\mathcal{J}) = (\rho\Delta P, -\Delta\mathcal{J}), \tag{27}$$

for $(P, \mathcal{J})$ above $C_+$ and $D_-$. Below the lines $C_-$ and $D_+$, the updates are opposite and the mean drift per update is leftward-upward, equal to

$$(-\rho\Delta P, \Delta\mathcal{J}), \tag{28}$$

for $(P, \mathcal{J})$ below $C_-$ and $D_+$. Depending on its initial condition and the exact set of updates drawn, this leads $(P, \mathcal{J})$ to reach either one of the two 'convergence corridors', between the lines $C_+$ and $D_+$, or between the lines $C_-$ and $D_-$ (see *Figure 9B*). In the triangular domain below the line $C_-$, the mean leftward-upward drift has an angle greater than the inclination of the $C_-$ line for $\rho\Delta P(1-q)>\Delta\mathcal{J}$ (i.e. $\beta<(1-q)\rho$). In this case, the mean drift does not ensure that the $(P, \mathcal{J})$ trajectory crosses the $C_-$ line. However, if $P$ becomes zero before crossing $C_-$, the positivity constraint on the rate, imposes that subsequent updates are strictly upward, until $(P, \mathcal{J})$ reaches $C_-$. Examples of such trajectories can be seen in *Appendix 1—figure 1C and D* (red lines).

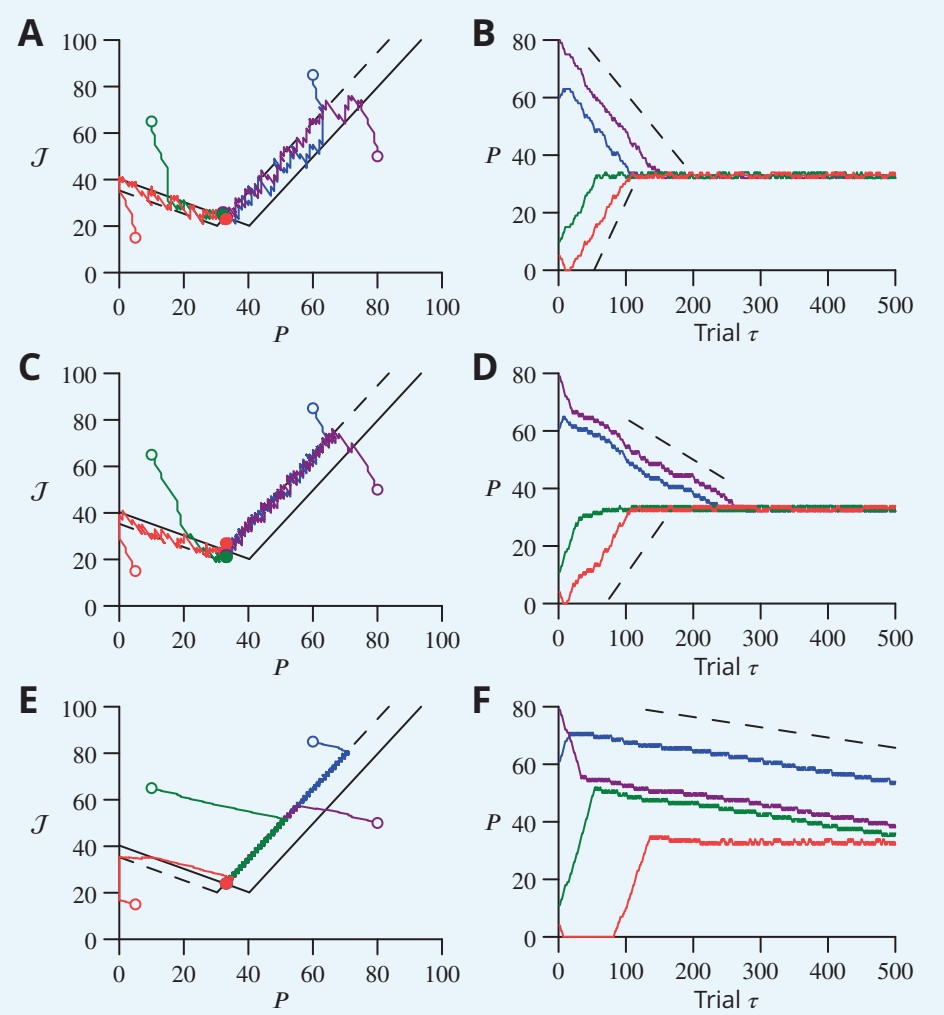

**Appendix 1—figure 1.** Reduced model: different cases of stochastic gradient learning are illustrated, for the parameters marked by *solid red circles* in **Figure 9C**, except for the case already shown in **Figure 9E and F**. (**A**) Dynamics in the $(P, \mathcal{J})$ plane for $\rho = 0.5$ and $\beta = 2$ ($A = 10, \Delta P = 1, \Delta \mathcal{J} = 2$). Trajectories (*solid lines*) from different initial conditions (*open circles*) are represented in the $(P, \mathcal{J})$ plane. The trajectories converge by oscillating around $C_+$ in the $C_+/D_+$ 'corridor' and around $D_-$ in the $C_-/D_-$ 'corridor'. Trajectory endings are marked by *filled circles*. (**B**) Time courses of $P(\tau)$ with corresponding colours. The slope of convergence (*dotted lines*) predicted by **Equation 32 and 33** agrees well with the observed convergence rate for $C_+$ while it is less accurate for $D_-$ when the assumption that the 'corridor' width is much greater than $\Delta P$ and $\Delta \mathcal{J}$ does not hold. (**C, D**) Same graphs for $\rho = 0.75$. The trajectories converge by oscillating around $C_+$ in the $C_+/D_+$ and around $C_-$ in the $C_-/D_-$ 'corridor'. (**E, F**) Same graphs for $\rho = 0.75$ and $\beta = 0.25$ (i.e. $A = 2, \Delta P = 1, \Delta \mathcal{J} = 0.25$). The trajectories converge by oscillating around $C_+$, which is the only attractive line.

DOI: https://doi.org/10.7554/eLife.31599.019

When a corridor is reached, in a second phase, $(P, \mathcal{J})$ follows a stochastic walk in the corridor.

A sufficient condition for convergence is that a single update cannot cross the two boundary lines of a corridor at once. For the $(C_+, D_+)$ corridor, the crossing by a single $\Delta P \Delta \mathcal{J}$ update provides the most stringent requirement,.

$$A > \Delta \mathcal{J}/(q+1) + \Delta P = \Delta \mathcal{J}[1/(q+1) + 1/\beta]. \tag{29}$$

The crossing of the $(C_-, D_-)$ corridor by a single $\Delta \mathcal{J}$ provides the other requirement

$$A(1-q) > \Delta\mathcal{J} \tag{30}$$

When these conditions are met, alternation between the two types of updates produces a mean downward drift of the error. This downward drift controls the convergence rate and depends on the relative size of the perturbation and the discrete $\Delta P$ and $\Delta\mathcal{J}$ modifications (in other words, the number of modifications that are needed to cross the convergence corridor). For simplicity, we consider the case where the perturbation $A$ is large compared to $\Delta P$ and $\Delta\mathcal{J}$, so that oscillations basically take place around one line of the corridor, as illustrated in **Figure 9E** and **Appendix 1—figure 1**. This amounts to being able to neglect the probability of crossing the other line of the corridor. The convergence rate then depends on the corridor line around which it takes place and can be obtained by noting that around a given corridor line, one of the two types of updates always has the same sign.

In the $(C_+, D_+)$ corridor, performing type $\Delta P\Delta\mathcal{J}$ updates for fraction $\rho$ of the steps and type $\Delta\mathcal{J}$ updates for the complementary fraction $(1-\rho)$ leads to the average displacement per step,

$$\rho(-\Delta P, \Delta\mathcal{J}) + (1-\rho)(0, -\Delta\mathcal{J}) = (-\rho\Delta P, (2\rho - 1)\Delta\mathcal{J}) \tag{31}$$

Thus, as summarised in **Figure 9C**

- For $\rho < \beta/[(q+1) + 2\beta]$ the average displacement leads to $D_+$ (**Figure 9E**). The average downward drift in the corridor can be obtained by noting that for a large $A$, the $(P, \mathcal{J})$ trajectory has a negligibly small probability of crossing the $C_+$ line. Therefore, all the type $\Delta P\Delta\mathcal{J}$ updates are of the same sign, of the form $(-\Delta P, \Delta\mathcal{J})$, and chosen with probability $\rho$. Since updates of type $\Delta\mathcal{J}$ do not change the value of $P$, $P$ approaches its target rate with a mean speed per step $V_c$,

$$D_+: V_c = -\rho\Delta P \tag{32}$$

A comparison between this computed drift and simulated trajectories is shown in **Figure 9F**.

- For $\rho > \beta/[(q+1) + 2\beta]$, the average displacement leads to $C_+$ (**Appendix 1—figure 1A,C,E**). In this case, all type $\Delta\mathcal{J}$ updates are of the same sign, of the form $(0, -\Delta\mathcal{J})$. In contrast, a fraction $f$ of type $\Delta P\Delta\mathcal{J}$ updates are of the form $(-\Delta P, \Delta\mathcal{J})$, while a fraction $(1-f)$ is of the opposite form $(\Delta P, -\Delta\mathcal{J})$ with $f$ to be determined. The average drift is thus $(1-\rho)(0, -\Delta\mathcal{J}) + \rho[f(-\Delta P, \Delta\mathcal{J}) + (1-f)(\Delta P, -\Delta\mathcal{J})] = (\rho(1-2f)\Delta P, [(\rho-1) + \rho(2f-1)]\Delta\mathcal{J})$. Requiring that this drift has slope $(1+q)$, like $C_+$, gives $2f - 1 = \beta(1-\rho)/[(1+q+\beta)\rho]$ and $f = [\beta + \rho(1+q)]/[2\rho(1+q+\beta)]$ (which indeed obeys $0 < f < 1$ in the parameter domain considered). Thus, $P$ approaches its target rate with a mean speed per step $V_c$,

$$C_+: V_c = -\frac{1-\rho}{1+q+\beta}\Delta\mathcal{J} \tag{33}$$

A comparison between this computed drift and simulated trajectories is shown in **Appendix 1—figure 1B,D,F**.

In the $(C_-, D_-)$ corridor, the average drift per step is opposite to the drift in the $(C_+, D_+)$ corridor (**Equation 31**). Thus,

- For $\beta > (1-q)$ and $\rho > \beta/(2\beta + q - 1)$ the average displacement leads to $C_-$ (**Appendix 1—figure 1A**). Updates of type $\Delta\mathcal{J}$ are always of the form $(0, \Delta\mathcal{J})$, while a fraction $f$ of type $\Delta P\Delta\mathcal{J}$ updates are of the form $(\Delta P, -\Delta\mathcal{J})$ and a fraction $(1-f)$ are of the opposite form $(-\Delta P, \Delta\mathcal{J})$. Again, since the average drift is along $C_-$ of slope $q-1$, one obtains $2f - 1 = \beta(1-\rho)/[(\beta + q - 1)\rho]$ or $f = [\beta + \rho(q-1)]/[2\rho(\beta + q - 1)]$ (which obeys $0 < f < 1$ in the parameter domain considered). The convergence speed $V_c$ is thus,

$$C_-: V_c = \frac{1-\rho}{\beta + q - 1}\Delta\mathcal{J} \tag{34}$$

A comparison between this computed drift and simulated trajectories is shown in **Appendix 1—figure 1B**.

- For $\beta < 1 - q$, or $\{\beta > 1, \rho < \beta(2\beta - 1)\}$, the complementary parameter domain, the drift in the corridor leads to $D_-$ (**Appendix 1—figure 1A,C**). The domain $\beta < (1 - q)\rho$ can be excluded, since when the point $(P, \mathcal{J})$ crosses $D_-$, the drift in the upper quadrant $(C_+, D_-)$ (**Equation 27**) tends to bring it to the other corridor $(C_+, D_+)$ (**Appendix 1—figure 1E**). Near the line $D_-$, type $\Delta P\Delta\mathcal{J}$ updates are always of the form $(\Delta P, -\Delta\mathcal{J})$. The convergence speed $V_c$ is thus

$$D_-: V_c = \rho\Delta P \tag{35}$$

A comparison between this computed drift and simulated trajectories is shown in **Figure 9F** and **Appendix 1—figure 1B**.

## Perceptron model and simulations

The architecture of the perceptron model is again the simplified circuit of **Figure 9A**, with $N_M = 1000$ mossy fibres projecting onto a single P-cell, with weights $w_i$, $i = 1, \ldots, N_M$ (which are all positive or zero). $N_P$ patterns are generated randomly. Activities of mossy fibres in different patterns are i.i.d. binary random variables $M_i^\mu$ with coding level $f$ (i.e. $M_i = 1$ with probability $f$ and 0 with probability $1 - f$; in the present simulations $f = 0.2$). The P-cell desired rates for the $N_P$ patterns are i.i.d. uniform variables $R^\mu$ from 0 to $P_{max}$ (=100 Hz).

In the case of the stochastic gradient descent with estimated global errors algorithm, at each trial a pattern $\mu$ is randomly drawn without replacement among the total $N_P$ and there is a probability $\rho$ of a perturbation of amplitude $A$. The output of the P-cell is

$$P^\mu = \left[\frac{1}{\sqrt{N_M}}\left(\sum_i w_i M_i^\mu - \theta N_M\right)\right]_+ + \eta A \tag{36}$$

where $\eta = 1$ with probability $\rho$ and $\eta = 0$ otherwise, which thus introduces a random perturbation of amplitude $A$ into the P-cell firing. The error is defined as $\mathcal{E} = |P^\mu - R^\mu|$. Comparison with previously obtained results for the capacity of this analog perceptron (**Clopath and Brunel, 2013**) motivates our choice of weights normalisation and the parameterisation of the threshold as $\theta = \frac{1}{2}P_{max}\left(\sqrt{\frac{f}{3(1-f)\gamma}} - \frac{1}{\sqrt{N_M}}\right)$, where $\gamma$ is a composite parameter reflecting the statistics of input and output firing, but here is equal to one.

An inferior olivary neurone receives the excitatory error signal but it also receives inhibitory inputs from the nucleo-olivary neurones driven by the mossy fibre inputs (which we have denoted $M$ above), with weights $v_i$. These are also plastic and sign-constrained. They represent the current estimated error. The net drive of the inferior olivary neurone is

$$IO = \mathcal{E} - \mathcal{I}, \text{ with } \mathcal{I} = \left[\frac{1}{\sqrt{N_M}}\left(\sum_i v_i M_i^\mu - \theta N_M\right) - qP^\mu\right]_+. \tag{37}$$

Here, the term $qP^\mu$ represents the specific inhibition of nucleo-olivary neurones by Purkinje cells and the term proportional to $\theta$ represents non-specific inhibition. In other words, $\frac{1}{\sqrt{N_M}}\left(\sum_i v_i M_i^\mu - \theta N_M\right)$ corresponds to $\mathcal{J}$ in **Equation 17**. For simplicity we assume the non-specific inhibition to be constant and the value of $\theta$ equal to that used in **Equation 36**.

The climbing fibre signal controlling plasticity is

$$c = \text{sign}(IO)$$

Weights are changed according to the following rule. Weights of $M \to P$ synapses active simultaneously with a perturbation are increased if $c$ is negative and decreased if it is positive. Weights of $M \to NO$ synapses are increased if $c$ is positive and decreased if not. Thus

$$w_i = [w_i - \alpha_w c \eta M_i^\mu]_+ \tag{38}$$

$$v_i = [v_i + \alpha_v c M_i^\mu]_+ \tag{39}$$

with the brackets indicating rectification (to impose the excitatory constraint).

The parameters of the simplified model of the previous subsection can be written as a function of those controlling the learning process in the present analog perceptron. The probability $\rho$ of the two types of updates (with and without perturbation) and the amplitude of perturbation $A$ are clearly identical in both models. In order to relate the previous $\Delta P$ and $\Delta \mathcal{J}$ to the present amplitude change of the weights $\alpha_w$ at mossy fibre–P-cell inputs and $\alpha_v$ for the indirect drive to the inferior olive from mossy fibres, we neglect the rectification constraints in **Equations 38 and 39** which is valid as long as synaptic weights are not very small (i.e. comparable to $\alpha_w$ or $\alpha_v$; see below for further discussion). Therefore, the weight modifications result in the changes $\Delta P$, of the perceptron firing rate, and $\Delta \mathcal{J}$, of the inhibitory input $\mathcal{I}$ to the olive,

$$\Delta P = \frac{1}{\sqrt{N_M}} \sum_i \alpha_w M_i^\mu = \alpha_w f \sqrt{N_M} \tag{40}$$

$$\Delta \mathcal{J} = \frac{1}{\sqrt{N_M}} \sum_i \alpha_v M_i^\mu = \alpha_v f \sqrt{N_M} \tag{41}$$

The convergence rate estimates of the previous section are compared with direct stochastic gradient descent learning simulations of the analog perceptrons in **Figure 10**. As shown in **Figure 10A**, for many single patterns, the convergence rate agrees well with the estimate **Equation 33**. However, for a fraction of the trajectories, the rate of convergence is about half of the prediction. This arises because these simulations constrained synaptic weights to be non-negative, which creates a significant fraction of synapses with negligible weights (**Brunel et al., 2004**) and produces a smaller effective step $\Delta P$ than estimated without taking this positivity constraint into account. For a larger number of patterns below the maximal learning capacity, the convergence rate per pattern is slower by a factor of $\approx 1.5 - 5$ for $N_P$ between $100 - 350$ (**Figure 10B**).

The SGDEGE algorithm is compared to the usual delta rule in **Figure 10**. For the delta rule, the patterns are presented sequentially. When the pattern $\mu$ is presented, the weights are changed according to the signed error $\mathcal{E}_s$ as,

$$w_i = [w_i - \alpha_w M_i^\mu \mathcal{E}_s]_+ \tag{42}$$

The error is defined by the distance (positive or negative) of the P-cell firing rate to the target rate, $\mathcal{E}_s = P^\mu - R^\mu$, and $P^\mu$ is given as before by **Equation 36** (with $A = 0$).

The convergence of the SGDEGE algorithm is considerably slower than that obtained for the delta rule (**Figure 10C**), while the relative slowing due to interference of multiple patterns is comparable for the delta rule and the proposed algorithm (compare panels B,C in **Figure 10**). The slower convergence rate of the SGDEGE algorithm arises from the use of update steps of fixed amplitude while the update is proportional to the error magnitude for the delta rule. A modified delta rule with constant amplitude updates would give a convergence rate comparable to the SGDEGE one.

