## [Decision Letter]

Thank you for submitting your article "Cerebellar learning using perturbations" for consideration by *eLife*. Your article has been reviewed by three peer reviewers, including Jennifer L Raymond as the Reviewing Editor and Reviewer #1, and the evaluation has been overseen by Eve Marder as the Senior Editor. The following individual involved in review of your submission has agreed to reveal their identity: Volker Steuber (Reviewer #3).

The reviewers have discussed the reviews with one another and the Reviewing Editor has drafted this decision to help you prepare a revised submission.

Note from the Senior Editor:

This manuscript does not comply to the intent of *eLife* papers. I thought about asking you to reconfigure prior to review, but thought I would wait until after the review because I would be asking you to do a considerable amount of rewriting/reorganization.

The issue: *eLife* is perfectly willing to publish long papers with many figures if the work merits. *eLife* is willing to publish two back-to back papers. What we will not do is emulate the journals that enforce a short format and cause authors to do what you did: write a paper with 20 pages and a second paper twice as long that is placed in supplemental but without which the first paper makes no sense. *eLife* wishes to publish a paper that makes sense. You will see that the reviewers were frustrated by having to flip back and forth between your two manuscripts.

So, in addition to the specifics of the review, I am asking to write the paper that is most transparent to the reader. Please look carefully at the *eLife* format and use it well. Some of your supplementary figures can be figure supplements. You can create a Mathematical Section if there are sections of math that are not necessary for the flow of the main story. But make sure that the reader is told the story in a simple way with enough detail for it to make sense.

Prepared by the Reviewing Editor:

Summary:

This manuscript presents a radical and intriguing new hypothesis about how the cerebellum implements learning. The authors propose a cerebellar learning algorithm that is based on stochastic gradient descent. The algorithm enables learning individual neuronal spike rates in the presence of global error signals and provides a potential solution for the credit assignment problem that commonly occurs in learning tasks. The implementation of stochastic gradient descent in the cerebellum is investigated with two different computational models: by simulating a network model of a cerebellar microzone, and by analysing a reduced version of this model that integrates the projection neurons in the cerebellar nuclei (CN) into the Purkinje cell (PC) pathway, and the nucleo-olivary neurons into a separate inhibitory pathway to the inferior olive (IO). The simulations and analyses show that such an implementation of cerebellar stochastic gradient descent can work in principle, but with a slower learning rate and slightly reduced capacity than classic perceptron learning. The suggested cerebellar implementation of stochastic gradient descent relies on a number of interesting assumptions that are in contrast to classic Marr-Albus type theories of cerebellar computation, such as (1) the presence of two different types of complex spikes (CSs), perturbation and error complex spikes (pCSs and eCSs), (2) the induction of parallel fibre (PF) long-term depression (LTD) in response to a pCS – eCS sequence, while long-term potentiation (LTP) is induced in response to pCS alone (both limited to active PF synapses), (3) the ability of pCSs to transiently increase the PC spike rate and elicit movements that can result in error signals, (4) the ability of nucleo-olivary input to extract the change in error by comparing a running average with the current error, based on plasticity between mossy fibres and nucleo-olivary neurons, and (5) the independence of nucleo-olivary inhibition on PC simple spike activity. Experimental support for assumption (2) is provided from in vitro recordings under conditions designed to approximate physiological conditions. Some of the other assumptions draw support from the experimental literature. The work that is presented in the article is novel, beautifully executed, with very important implications for cerebellar learning theories, and the broader learning and memory field. The experimental confirmation of the proposed novel learning rules is an exemplar of a systems biology approach to computational neuroscience; experimental tests of theoretical predictions are in fact much too rare and should be conducted more often. However, the reviewers had some substantial concerns.

Essential revisions:

1) A key control is missing from the slice experiments.

The central claim is that associative LTD requires that parallel fiber activation be accompanied by a "perturbation" complex spike (driven by 2 climbing fiber stimuli) and followed ~100 ms later by an "error" complex spike (driven by 4 climbing fiber stimuli). To establish that this particular pattern is necessary, control experiments are needed whereby the parallel fibers are paired with *the same number* of climbing fiber stimuli but without the perturbation-error patterning, i.e., that it is not sufficient to simply have the same number of climbing fiber pulses anywhere within the typical LTD pairing window of ~0-100 ms. Does LTD occur if the parallel fiber activation is simply paired with a train of 6 climbing fiber stimuli at 400 Hz, either at 0 ms or 100 ms delay? If not, then this would greatly strengthen the claim that the climbing fiber "perturbation" creates an eligibility trace that interacts with a later climbing fiber error signal. But if the LTD can be induced with parallel fiber plus a train of 6 climbing fiber stimuli, then this would look more like old-fashioned cerebellar LTD. The number 6 climbing fiber stimuli was used here, because that is the only condition that yielded LTD in the experiments that were presented.

2) The clarity of the writing is in need of improvement.

The brief main text contains a lot of information, with a number of references to the supplementary material, and is not very easy to understand on its own. To understand the manuscript, one has to read all of the supplementary material (which in effect provides an extended version of the article). It would be very helpful to try to expand on a few things in the main text. particularly the network model and the justification of some of the assumptions, potentially at the expense of other content. For example, the authors might consider leaving out the comparison with perceptron learning and focus on and expand the (very important) cerebellar content. Another example is in the Introduction the authors frame the problem by stating that in this study, 'we shall simulate learning tasks that the Marr-Albus-Ito theory would be incapable of optimising'. Ideally the authors would follow this with a description of such a motor learning context including how current theory is insufficient.

3) The strong claims that the slice experiments were done under physiological conditions whereas previous LTD experiments were not should be toned down.

All slices are not physiological in many ways. The current slice experiments were done in ways that differ from many previous cerebellar LTD experiments in several parameters (including superposition of the LTD induction protocols on a basal 0.5 Hz climbing fiber stimulation rate, which is not even mentioned in the main text), and these parameters are not varied independently, so we have no idea which choices matter or not. There are also some curious technical details in these experiments, such as the absence of any typical concentration (e.g. 1-3 mM) of a conventional calcium buffer (e.g. EGTA) in the pipette solution. The authors report calcium measurements of ~120 nM in the Materials and methods, but how are the unavoidable trace calcium concentrations found in most water sources buffered in their pipette solution recipe?

Although there is some merit to the authors' claims about the principles guiding their choices of conditions, the confidence with which the authors assert that their choices are physiological whereas standard protocols are exceeds the evidence regarding what is physiological. We really have no idea what the physiological pattern of parallel fiber activation is in vivo (clustered vs. spatially dispersed), and I am not even convinced we know exactly what the extracellular calcium concentration is in the cerebellar molecular layer.

Making different experimental choices than the "standard" protocols, and asking how that affects experimental outcomes is great for the field. But making different, and not much less arbitrary sets of choices, and then making big claims that these choices have a critical effect on the results without doing the necessary direct comparisons across choices of experimental conditions does not increase clarity.

4) One of the assumptions that got lost in the short version of the article is (5) from above – the inability of PCs to affect nucleo-olivary neurons in the CN, except through the effect of eCSs on mossy fibre – CN neuron plasticity (subsection “Simulation methods” of the supplementary material – in the reduced model this is reflected by making I independent of P). This is not supported by the literature, see e.g. Chan-Palay, 1977, De Zeeuw, 1988, Teune, 1998, Najac and Raman, 2015). The modulation of nucleo-olivary input by PC activity should be detrimental to the stochastic gradient descent; could the authors show, by running additional simulations (e.g. with varying PC weights), how detrimental this would be?

5) pCSs are assumed to increase the PC spike rate and elicit movements. It should also be mentioned that a CS also results in a temporary decrease of the PC spike rate, via CS induced pauses. More importantly, it is worrisome that the downstream effects of a pCS and eCS are treated differently in the models. Should a pCS not affect the nucleo-olivary neurons if an eCS does so? Furthermore, should an eCS not also result in a movement, similar to a pCS, which then would result in another error signal, etc.? This should be discussed in the main text.

6) *eLife* does not support the kind of large supplemental article that many journals do. Much of the relevant information should be brought into the body of the paper. If necessary, the authors can provide a mathematical Appendix, or an extended Materials and methods section.

[Editors' note: further revisions were requested prior to acceptance, as described below.]

Thank you for resubmitting your work entitled "Cerebellar learning using perturbations" for further consideration at *eLife*. Your revised article has been favorably evaluated by Eve Marder as the Senior Editor, and three reviewers, one of whom is a member of our Board of Reviewing Editors.

This manuscript presents a radical and intriguing new hypothesis about how the cerebellum implements learning.

The authors propose a cerebellar learning algorithm inspired by machine learning approaches, which uses a perturbationerror signal sequence carried by the cerebellar climbing fibers to guide the induction of plasticity. This approach provides a potential solution to a common credit assignment problem associated with using a global error signal to train a network.

The model relies on a number of interesting assumptions that are in contrast to the classic Marr-Albus-Ito theory of cerebellar learning and previous observations of synaptic plasticity in cerebellar slice preparations. In particular, the model predicts that associative long-term depression (LTD) of the parallel fiber-Purkinje cell synapses will only occur when parallel fiber activity is paired with a sequence of two, closely spaced 'perturbation' and 'error' complex spikes in the Purkinje cell, in contrast to a large body of previous work, which reported that LTD could be induced by appropriately timed pairings of parallel fiber activity with a single climbing fiber-driven complex spike. Using slice conditions that differ in a number of ways from typical cerebellar slice experiments, the authors found properties of LTD at the parallel fiber-Purkinje cell synapses that matched the predictions of their model. The revised manuscript includes an additional, key control experiment, which controls for the total number of climbing fiber stimuli delivered during induction of plasticity.

This is innovative and carefully executed work, and should generate considerable interest in the cerebellar field as well as the broader learning and memory field. The reviewers were skeptical about whether the algorithm proposed by the authors is actually the mechanism of cerebellum-dependent learning. Nevertheless, they were enthusiastic about the novelty of the work, and its potential to stimulate debate and additional study.

There was just one significant issue related to terminology, which the reviewers thought should be addressed. Specifically, use of the term "stochastic gradient decent" in the manuscript does not seem to agree with the general usage of this term and, in particular, with how it is used in the machine learning community. In general usage, the term means modifying connections on the basis of a gradient computed from randomly chosen examples. A more appropriate choice in the current case might be "learning through reinforced weight perturbation."

---

## [Author Response]

Essential revisions:1) A key control is missing from the slice experiments.The central claim is that associative LTD requires that parallel fiber activation be accompanied by a "perturbation" complex spike (driven by 2 climbing fiber stimuli) and followed ~100 ms later by an "error" complex spike (driven by 4 climbing fiber stimuli). To establish that this particular pattern is necessary, control experiments are needed whereby the parallel fibers are paired with the same number of climbing fiber stimuli but without the perturbation-error patterning, i.e., that it is not sufficient to simply have the same number of climbing fiber pulses anywhere within the typical LTD pairing window of ~0-100 ms. Does LTD occur if the parallel fiber activation is simply paired with a train of 6 climbing fiber stimuli at 400 Hz, either at 0 ms or 100 ms delay? If not, then this would greatly strengthen the claim that the climbing fiber "perturbation" creates an eligibility trace that interacts with a later climbing fiber error signal. But if the LTD can be induced with parallel fiber plus a train of 6 climbing fiber stimuli, then this would look more like old-fashioned cerebellar LTD. The number 6 climbing fiber stimuli was used here, because that is the only condition that yielded LTD in the experiments that were presented.

A brief clarification is in order. Although we only tested our theory with a 100ms interval between pCS and eCS, we expect that a wide range of such intervals (say 50–300ms) should induce LTD, because we don’t expect perturbations always to occur at the same point during a movement and because eCSs appear to arise within a fairly broad time window after a movement. Furthermore, our theory predicts specific plasticity outcomes of LTP when a pCS alone (GP_) occurs and LTD when there is a pCS then eCS sequence (GPE), as observed. The theory makes no prediction about the underlying molecular mechanism or about activity patterns that do not arise in vivo. In particular, a mechanism in which the two CSs produced LTD via simple summation of additional calcium influx compared to a single CS would be entirely compatible with the theory; this is addressed in the Discussion (subsection “Implications for studies of synaptic plasticity”, fifth paragraph). In other words, our theory predicts outcomes and does not require some molecular signalling nonlinearity.

Nevertheless, we understand the interest in determining whether different stimulus patterns with a similar overall intensity produce different outcomes. In response to this request, we applied a burst of 6 CF stimuli (although we note that such events are anyway likely to be very rare in vivo), the same number as occurring in the GPE protocol, synchronously with the granule cell input. A modest LTP resulted, which was clearly distinct from the LTD previously observed with GPE. These data have been included as Figure 6—figure supplement 1.

2) The clarity of the writing is in need of improvement.The brief main text contains a lot of information, with a number of references to the supplementary material, and is not very easy to understand on its own. To understand the manuscript, one has to read all of the supplementary material (which in effect provides an extended version of the article). It would be very helpful to try to expand on a few things in the main text. particularly the network model and the justification of some of the assumptions, potentially at the expense of other content. For example, the authors might consider leaving out the comparison with perceptron learning and focus on and expand the (very important) cerebellar content. Another example is in the Introduction the authors frame the problem by stating that in this study, 'we shall simulate learning tasks that the Marr-Albus-Ito theory would be incapable of optimising'. Ideally the authors would follow this with a description of such a motor learning context including how current theory is insufficient.

In order to produce a unified, complete manuscript, we decided to use as starting point the previous Supplementary Information, which contained much of the additional information requested in this point and also in point 6 (“Much of the relevant information should be brought into the body of the paper”). However, we have also made major changes to it to clarify and compress the presentation. We justify the present manuscript format as follows and also list the principal changes. The main text (excluding Materials and methods, References and Appendix) in the LaTeX format is just over 27 pages, including 11 figures.

- We have chosen to retain the condensed Introduction to the most relevant aspects of cerebellar physiology (first three paragraphs and Figure 1). This is of course superfluous for expert cerebellar physiologists, but we hope and aim to attract readers working on other brain areas and we feel this Introduction will be necessary for them.

- A large amount of introductory analytical material has been replaced by a more direct subsection (“Requirements for cerebellar learning”) that we hope exposes the fundamental limitations of current theory. We have performed some simulations (Figure 2), included in this section, to demonstrate more directly how current theory fails, as requested.

- The network simulations have been included in the manuscript, as requested (subsection “Simulations”).

- The new data have been included (described below).

- We have now included in all our analyses the Purkinje cell–nucleoolivary connection. Its influence is detailed below.

- A short mathematical section establishing the key convergence and capacity arguments has been retained, while more detailed analysis appears in an appendix. This analysis is novel. The “perceptron” analysis is integral to comparing the capacity of our algorithm with the theoretical maximum.

- The Discussion is essentially that from the previous supplementary information. This included information relevant to mechanisms for disambiguating pCS and eCS, requested by the reviewers. We have retained a brief section of what we believe to be an absolutely key question regarding climbing fibre modalities.

- Following the ‘Transparent reporting’ form, a more complete statistical analysis is described in the Materials and methods (subsection “Analysis”).

Given the very extensive changes to the manuscript and because we used the LaTeX template for the manuscript, it would be technically difficult and anyway of little utility to offer a manuscript version with all changes marked up. We apologise for this.

3) The strong claims that the slice experiments were done under physiological conditions whereas previous LTD experiments were not should be toned down.All slices are not physiological in many ways. The current slice experiments were done in ways that differ from many previous cerebellar LTD experiments in several parameters (including superposition of the LTD induction protocols on a basal 0.5 Hz climbing fiber stimulation rate, which is not even mentioned in the main text), and these parameters are not varied independently, so we have no idea which choices matter or not. There are also some curious technical details in these experiments, such as the absence of any typical concentration (e.g. 1-3 mM) of a conventional calcium buffer (e.g. EGTA) in the pipette solution. The authors report calcium measurements of ~120 nM in the Materials and methods, but how are the unavoidable trace calcium concentrations found in most water sources buffered in their pipette solution recipe?Although there is some merit to the authors' claims about the principles guiding their choices of conditions, the confidence with which the authors assert that their choices are physiological whereas standard protocols are exceeds the evidence regarding what is physiological. We really have no idea what the physiological pattern of parallel fiber activation is in vivo (clustered vs. spatially dispersed), and I am not even convinced we know exactly what the extracellular calcium concentration is in the cerebellar molecular layer.Making different experimental choices than the "standard" protocols, and asking how that affects experimental outcomes is great for the field. But making different, and not much less arbitrary sets of choices, and then making big claims that these choices have a critical effect on the results without doing the necessary direct comparisons across choices of experimental conditions does not increase clarity.

We have made several changes throughout the text in an attempt to avoid appearing over confident of our physiological conditions. Nevertheless, they were designed with the specific aim of being as physiological as possible and, with all due respect, we cannot agree that they are”not much less arbitrary” than those of previous work. Thus, there can be no disagreement that keeping inhibition intact is physiological. Similarly, although we also wish that the precise extracellular calcium concentration had been more accurately determined, measurements do exist in the range we use, while none approach the 2–2.5mM previously used. We have added a reference to a measurement in the cerebellar molecular layer (Nicholson et al., 1978), which actually reports an even lower concentration of 1.2mM. In the presence of any kind of spatiotemporal randomness in the activation of mossy fibres, the highly compact bundles seen with molecular layer stimulation are surely extremely unlikely to arise. Similarly, continuous stimulation of the climbing fibre (now mentioned in the third paragraph of the subsection “Synaptic plasticity under physiological conditions”) also obviously resembles the in vivo situation.

The related changes in the text are to be found in the Abstract, subsections “Synaptic plasticity under physiological conditions”, “Implications for studies of synaptic plasticity” and “Implications for studies of synaptic plasticity”.

The reviewers are perfectly correct in pointing out the necessity of buffering contaminant calcium in the pipette solution. Our solution did contain 100µM EGTA. Unfortunately this was omitted from the recipe. The Materials and methods have been corrected (subsection “Electrophysiology”). Thank you for pointing this out. Note that Purkinje cells are thought to have very strong endogenous intracellular buffering (Fierro and Llano, 1996, “High endogenous calcium buffering in Purkinje cells from rat cerebellar slices”), so the precise concentrations of added buffer may be less significant in these cells than in some others.

4) One of the assumptions that got lost in the short version of the article is (5) from above – the inability of PCs to affect nucleo-olivary neurons in the CN, except through the effect of eCSs on mossy fibre – CN neuron plasticity (subsection “Simulation methods” of the supplementary material – in the reduced model this is reflected by making I independent of P). This is not supported by the literature, see e.g. Chan-Palay, 1977, De Zeeuw, 1988, Teune, 1998, Najac and Raman, 2015). The modulation of nucleo-olivary input by PC activity should be detrimental to the stochastic gradient descent; could the authors show, by running additional simulations (e.g. with varying PC weights), how detrimental this would be?

We have now included an active Purkinje cell–nucleo-olivary connection throughout all of the simulations and theoretical analyses in the manuscript. The upshot of including the PC-NO connection is that it only degrades learning under some conditions. However, if a constraint relating to the perturbation is satisfied, there is no effect on learning convergence or capacity. The transmission of the perturbation via the projection neurones affects the movement and the error, while the transmission via the nucleo-olivary neurones to the inferior olive alters the error estimate. The former effect must remain larger than the latter for the algorithm to function correctly. In our models, this constraint can be expressed as a relation between the strengths of the PCPN and PC-NO connections, *q* in our nomenclature. This is most clearly demonstrated in Figure 10D.

5) pCSs are assumed to increase the PC spike rate and elicit movements. It should also be mentioned that a CS also results in a temporary decrease of the PC spike rate, via CS induced pauses. More importantly, it is worrisome that the downstream effects of a pCS and eCS are treated differently in the models. Should a pCS not affect the nucleo-olivary neurons if an eCS does so? Furthermore, should an eCS not also result in a movement, similar to a pCS, which then would result in another error signal, etc.? This should be discussed in the main text.

Complex spikes do induce pauses in Purkinje cells. However, as mentioned in the text (subsection “The complex spike as trial and error”, fifth paragraph) the recordings of Bengtsson and Jorntell (2011) indicate that complex spikes in Purkinje cells produce net inhibition rather than net excitation in nuclear projection neurones. For this reason we have considered the complex spike to be excitatory. In the Discussion we suggest that granule cell synapses active during the complex spike pause might be depressed rather than undergoing the LTP observed with exact synchrony (subsection “Implications for studies of synaptic plasticity”, last paragraph), but this has not been tested.

In both simulations and theoretical analysis the pCS is transmitted to the nucleo-olivary neurones (see point 4).

Regarding the possible confusion between pCS and eCS (from the viewpoint of a Purkinje cell), we have expanded the Discussion (originally in the supplementary information) on this point, starting in the third paragraph of the subsection “Possible extensions to the algorithm”. In brief, Purkinje cells may be able to distinguish the two types of complex spike and produce different plasticity – ideally zero change for an isolated eCS. Although we might expect an eCS to affect a movement, all such perturbations are expected to be small and the eCS is, by definition, assumed to arise after a movement and its evaluation. A small movement change outside of the context of a movement may not be serious.

6) eLife does not support the kind of large supplemental article that many journals do. Much of the relevant information should be brought into the body of the paper. If necessary, the authors can provide a mathematical Appendix, or an Extended Materials and methods section.

See the above description (point 2) of the reconfiguration of the manuscript. We have followed the suggestion of including a mathematical appendix.

[Editors' note: further revisions were requested prior to acceptance, as described below.]

[…] This is innovative and carefully executed work, and should generate considerable interest in the cerebellar field as well as the broader learning and memory field. The reviewers were skeptical about whether the algorithm proposed by the authors is actually the mechanism of cerebellum-dependent learning. Nevertheless, they were enthusiastic about the novelty of the work, and its potential to stimulate debate and additional study.There was just one significant issue related to terminology, which the reviewers thought should be addressed. Specifically, use of the term "stochastic gradient decent" in the manuscript does not seem to agree with the general usage of this term and, in particular, with how it is used in the machine learning community. In general usage, the term means modifying connections on the basis of a gradient computed from randomly chosen examples. A more appropriate choice in the current case might be "learning through reinforced weight perturbation."

It was requested that we change our terminology to avoid potential confusion with current practice in the machine learning community. For the reasons outlined next, we believe that the best approach will be to retain our terminology but to add to the Introduction a clarification for readers from this community.

Firstly, stochastic gradient descent can reasonably be taken to describe a broad class of algorithms of which ours and that used in machine learning are specific examples: they both descend gradients and include an element of stochasticity. Secondly, use of the term stochastic gradient descent to describe learning with perturbations of synaptic weights or of neuronal output has a long pedigree in neuroscience. For instance, the term is used in Doya and Sejnowski (1998) and also in Seung (2003). Thirdly, the detailed description of our algorithm should make its nature abundantly clear to interested readers. Finally, the reviewer’s specific suggestion is potentially misleading because weights are not perturbed in our algorithm (it might be described as involving perturbations of neuronal output).

We have added a brief explanation in the Introduction (sixth paragraph), distinguishing the two types of stochastic gradient descent.